# MERTK mediated novel site Akt phosphorylation alleviates SAV1 suppression

Yao Jiang[1,2,3], Yanqiong Zhang [2,3], Janet Y. Leung[2,4], Cheng Fan[2,5,6], Konstantin I. Popov[3], Siyuan Su[2,3], Jiayi Qian[2,3], Xiaodong Wang[2,7], Alisha Holtzhausen [2,8], Eric Ubil[2,8], Yang Xiang[9], Ian Davis[2,5,10], Nikolay V. Dokholyan [2,3,11], Gang Wu[1], Charles M. Perou[2,5,6], William Y. Kim[2,4,5,8], H. Shelton Earp[2,8] & Pengda Liu[2,3]

Akt plays indispensable roles in cell proliferation, survival and metabolism. Mechanisms underlying posttranslational modification-mediated Akt activation have been extensively studied yet the Akt interactome is less understood. Here, we report that SAV1, a Hippo signaling component, inhibits Akt, a function independent of its role in Hippo signaling. Binding to a proline-tyrosine motif in the Akt-PH domain, SAV1 suppresses Akt activation by blocking Akt's movement to plasma membrane. We further identify cancer-associated SAV1 mutations with impaired ability to bind Akt, leading to Akt hyperactivation. We also determine that MERTK phosphorylates Akt1-Y26, releasing SAV1 binding and allowing Akt responsiveness to canonical PI-3K pathway activation. This work provides a mechanism underlying MERTK-mediated Akt activation and survival signaling in kidney cancer. Akt activation drives oncogenesis and therapeutic resistance; this mechanism of Akt regulation by MERTK/SAV1 provides yet another complexity in an extensively studied pathway, and may yield prognostic information and therapeutic targets.

[1] Cancer Center, Union Hospital, Tongji Medical College, Huazhong University of Science and Technology, Wuhan 430022, China. [2] Lineberger Comprehensive Cancer Center, The University of North Carolina at Chapel Hill, Chapel Hill, NC 27599, USA. [3] Department of Biochemistry and Biophysics, The University of North Carolina at Chapel Hill, Chapel Hill, NC 27599, USA. [4] Department of Medicine, The University of North Carolina at Chapel Hill, Chapel Hill, NC 27599, USA. [5] Department of Genetics, The University of North Carolina at Chapel Hill, Chapel Hill, NC 27599, USA. [6] Department of Pathology and Laboratory Medicine, The University of North Carolina at Chapel Hill, Chapel Hill, NC 27599, USA. [7] Center for Integrative Chemical Biology and Drug Discovery, Division of Chemical Biology and Medicinal Chemistry, Eshelman School of Pharmacy, The University of North Carolina at Chapel Hill, Chapel Hill, NC 27599, USA. [8] Department of Medicine and Pharmacology, The University of North Carolina at Chapel Hill, Chapel Hill, NC 27599, USA. [9] Abclonal Technology, 86 Cummings Park Drive, Woburn, MA 01801, USA. [10] Department of Pediatrics, The University of North Carolina at Chapel Hill, Chapel Hill, NC 27599, USA. [11] Present address: Departments of Pharmacology, Biochemistry and Molecular Biology, Pennsylvania State College of Medicine, Hershey, PA 17033, USA. These authors contributed equally: Yao Jiang, Yanqiong Zhang. Correspondence and requests for materials should be addressed to P.L. (email: pengda_liu@med.unc.edu)

Advances in genomics, gene function annotation, and systems biology have revealed pathways in normal cells that are deranged in cancer; this knowledge serves as a blueprint for targeted cancer therapy[1]. Protein–protein interactions (PPI), often governed by posttranslational modifications, play an essential role in integrating proteins into signal transduction pathways and networks. Although more than 650,000 PPIs have been predicted by large-scale proteomics approaches[2–4], so far only a few agents such as venetoclax approved to treat patients with chronic lymphocytic leukemia (CLL) directly target PPIs[5]. This is in large part due to the limited association between PPIs and pathophysiological functions.

The oncogenic protein kinase Akt plays essential roles in regulating cell proliferation, survival, metabolism, and genome stability[6]. Hyperactivation of Akt has been observed virtually in all solid tumors[7,8] and has been shown to drive tumorigenesis in different cancer settings using a variety of murine models[9]. In addition to *Akt* gene amplification and mutation[10], various extracellular signals drive posttranslational modifications of Akt in normal and neoplastic cells, controlling Akt activation and oncogenicity, including phosphorylation[11–13], hydroxylation[14], acetylation[15], ubiquitination[16,17], and others. Accordingly, inhibitors targeting these modifying enzymes have been developed and examined clinically, currently with limited success[18]. In contrast to well-characterized Akt posttranslational modifications identified in the past[19], our knowledge about how non-enzymatic Akt binding proteins modulate Akt activity in cancer is limited, and whether Akt PPIs can be exploited for cancer therapy remains to be determined. With rapidly developing techniques to make targeting oncogenic PPIs feasible[20], these findings would shed light into both Akt biology and cancer therapeutics. Here, we identify SAV1 as an Akt endogenous inhibitor and SAV1-mediated Akt suppression can be released by either MERTK-mediated Akt1-Y26 phosphorylation or by cancerous SAV1 mutations with deficiencies in binding Akt. Thus, our results suggest that both SAV1 and MERTK contribute to Akt activity regulations, and SAV1 is a critical component for MERTK inhibitor-mediated suppression of Akt activation in renal cell carcinoma.

## Results

**SAV1 is an endogenous Akt inhibitor: SAV1 binds and suppresses Akt activation.** Since the Akt signaling regulates cell size[21], whereas the Hippo signaling controls cell growth by modulating organ size[22], we hypothesized that these pathways may be coordinated. Given that WW-domains in various Hippo signaling pathway members, such as SAV1, LATS1, YAP, and TAZ, mediate Hippo signal transduction[23], we investigated their interactions with Akt finding only one of these key WW-domain containing Hippo components, SAV1, but not others (such as YAP and TAZ), specifically bound Akt1 in cells (Fig. 1a, 1b). Furthermore, this interaction was mediated by the SAV1-WW domain (Supplementary Fig. 1a to 1c) and the Akt1-PH domain (Supplementary Fig. 1d and 1e). Given that the "PxY" motif is a specific WW-domain binding motif[24], we identified a "$P_{24}R_{25}Y_{26}$" motif in the Akt1 PH domain (Fig. 1c) that is both evolutionarily conserved and present in all three Akt isoforms necessary for this interaction (Fig. 1d and Supplementary Fig. 1f). A structural simulation using available structures for the Akt1-PH domain and SAV1 suggests that both $P_{24}$ and $Y_{26}$ residues reside on the interaction surface between SAV1 and Akt1 (Supplementary Fig. 1g). Consistent with the notion that these residues are critical in mediating Akt1 interaction with SAV1, an Akt1-P24A mutant significantly reduced Akt1 binding to SAV1 (Fig. 1e and Supplementary Fig. 1h and 1i). In contrast, an Akt1-Y26F mutation

dramatically enhanced SAV1 binding (Fig. 1e and Supplementary Fig. 1h and 1i), presumably due to strengthened molecular interaction between SAV1 and Akt1 (Supplementary Fig. 1j and 1k). The fact that Akt1-Y26F is a tyrosine phospho-deficient mutation suggests that phosphorylation of the Akt1-Y26 residue would antagonize SAV1 binding to the Akt1-PH domain. In support of this hypothesis, we observed that Y26-phosphorylated Akt1-PH peptides were deficient in binding SAV1, as compared to non-phosphorylated peptide controls (Fig. 1f). Importantly, the Akt1-Y26F mutation dramatically reduced Akt-pT308 signals in cells (Fig. 1g and Supplementary Fig. 1l) and attenuated Akt activity in vitro (Supplementary Fig. 1m), whereas Akt1-P24A exhibited enhanced Akt-pT308 (Supplementary Fig. 1l). These data suggest that SAV1 binds directly with the Akt-PH domain to inhibit Akt activity.

Consistent with SAV1 being an Akt activity suppressor, kidney-specific *SAV1* deletion in mice[25] led to elevated Akt-pT308 (Fig. 1h). Moreover, reduced SAV1 expression was observed in renal cell carcinoma (RCC) patients (Supplementary Fig. 2a). In most pairs of clear cell renal cell carcinoma (ccRCC) patient samples we analyzed, reduced SAV1 expression correlated with high Akt-pT308 signals (Supplementary Fig. 2b). Moreover, consistent with previous reports that chromosomal *14q* (the location of *SAV1*) loss was observed in high-grade ccRCC and neuroblastoma[26,27], we detected a moderate inverse correlation between SAV1 expression and Akt-pT308 levels in a cohort of RCC patients with stage IV disease (Supplementary Fig. 2c), but not earlier stages (Supplementary Fig. 2d to 2f). Consistently, phosphorylation of an Akt substrate, GSK3β (Supplementary Fig. 2g) also moderately inversely correlated with reduced SAV1 expression in stage IV KIRC dataset, while the association of Akt-pS473 (Supplementary Fig. 2h) or Akt upstream activating enzymes including mTOR (Supplementary Fig. 2i) and PDK1 (Supplementary Fig. 2j) with SAV1 expression were not noticeably correlated. Notably, no such correlation was observed in KIRP dataset (Supplementary Fig. 2k to 2o). Interestingly, either low SAV1 expression (Supplementary Fig. 2p and 2q) or high Akt-pT308 levels (Supplementary Fig. 2r and 2s) predicted worse prognosis and survival in RCC patients. These observations suggest that deficiency in SAV1 may facilitate RCC development in part through promoting Akt activation in RCC patients. This also suggests that SAV1 may be an independent marker distinct from the canonical Akt control mechanisms, given that *SAV1* genetic alternations are not coincident with genetic alternations in *PIK3CA* or *PTEN* (Supplementary Fig. 2t). These data support that SAV1 may be a *bona fide* Akt suppressor independent of "classical" Akt regulatory pathways.

Consistent with a previous report[28], SAV1 expression was not detectable in two ccRCC cell lines, 786-O and ACHN (Fig. 1i). Depletion of endogenous SAV1 in SAV1-expressing RCC cell lines (Fig. 1i), including RCC4, UMRC6, and Caki-1, resulted in dramatically increased Akt-pT308 (Fig. 1j, k and Supplementary Fig. 3a) and enhanced cell growth (Fig. 1l, m). In contrast, ectopic expression of SAV1 in both SAV1-deficient (Supplementary Fig. 3b) and SAV1-proficient RCC (Supplementary Fig. 3c and 3d) and in non-RCC cells (Supplementary Fig. 3e to 3g) led to attenuated Akt-pT308 levels, which was predominately mediated by the SAV1-WW domain (Supplementary Fig. 3h and 3i). Notably, no significant changes of PTEN expression were observed upon manipulating SAV1 expression (Fig. 1h, j, l and Supplementary Fig. 3c to 3f)[29]. Our observed SAV1-mediated suppression of Akt activation may not function through MST1, given that multiple upstream signals in addition to MST1/SAV1 have been identified that activate LATS1/2 to suppress YAP/TAZ[30,31], which could compensate for SAV1 deficiency in regulating YAP activity. This is further supported by our

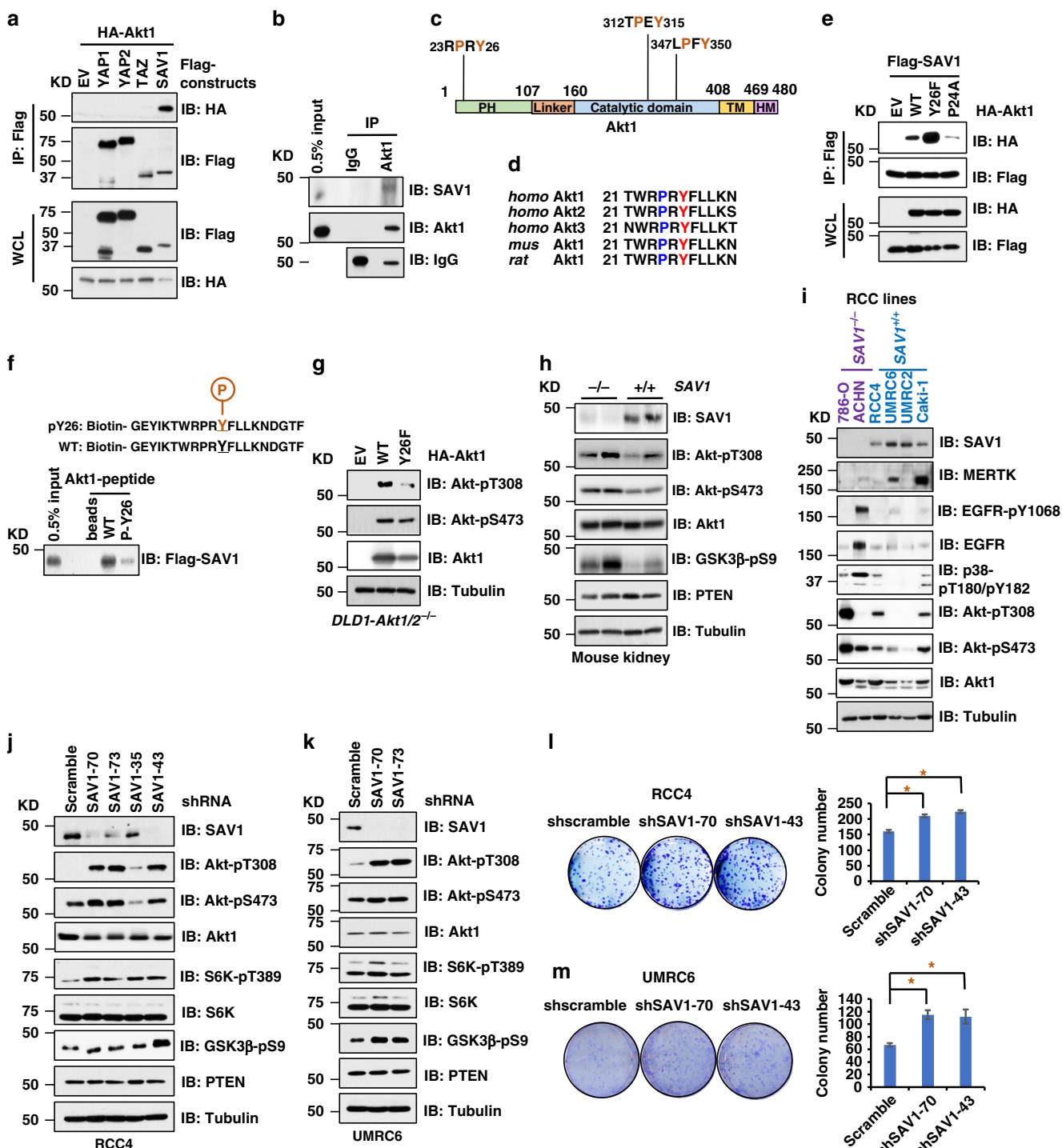

**Fig. 1** SAV1 binds and suppresses Akt activity. **a** Immunoblot (IB) analysis of whole cell lysates (WCL) and Flag-immunoprecipitates (IP) derived from HEK293 cells transfected with indicated DNA constructs. **b** IB analysis of WCL and endogenous Akt1-IP derived from RCC4 cells. **c** A cartoon illustration of the distribution of PxY motifs in Akt1. **d** Protein sequence alignment the "PxY" motif located in the PH domain of Akt isoforms. **e** IB analysis of WCL and Flag-IP derived from HEK293 cells transfected with indicated DNA constructs. **f** IB analysis of indicated Akt1 peptide pulldown products. **g** IB analysis of WCL derived from *DLD1-Akt1/2*−/− cells transfected with indicated HA-Akt1 constructs. **h** IB analysis of WCL derived from *WT* and *SAV1*−/− mouse kidney tissues. **i** IB analysis of WCL derived from indicated RCC lines. **j, k** RCC4 (**j**) and UMRC6 (**k**) cells were depleted of endogenous SAV1 by lentiviral shRNAs against SAV1 and harvested 72-h post puromycin selection (1 μg/ml) for IB analysis. **l, m** 600 cells resulted from (**j**) and (**k**) were subjected to colony formation assays

observation that in 293A cells, genetic deletion of neither *MST1/2* nor *LATS1/2* impaired Akt phosphorylation (Supplementary Fig. 3j). Although both Akt1 and MST1 bind SAV1, they did not appear to compete for SAV1 binding in cells (Supplementary

Fig. 3k and 3l), suggesting that SAV1 may not be a rate-limiting factor in cells for Hippo and Akt regulation. Interestingly, an inverse correlation between SAV1 expression and Akt-pT308 was observed in a panel of SAV1-expressing RCC lines

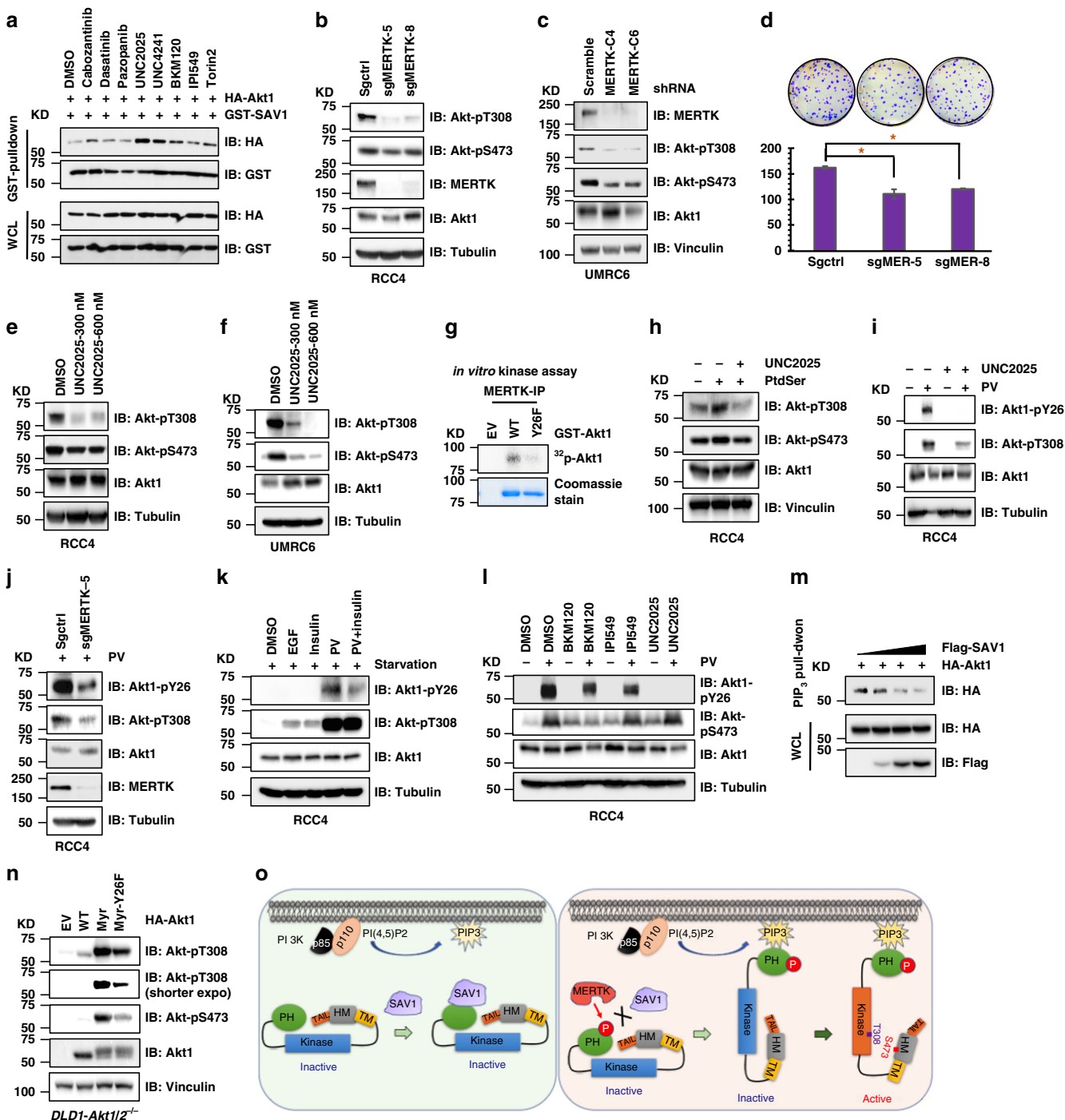

**Fig. 2** MERTK phosphorylates and activates Akt by releasing SAV1 binding. **a** IB analysis of WCL and GST-pulldown products derived from HEK293 cells transfected with CMV-GST-SAV1 and HA-Akt1 constructs and treated with indicated kinase inhibitors for 12 h before cell collection. Doses for inhibitors used as following: cabozantinib (c-Met and VEGFR2 inhibitor, 100 nM), dasatinib (Src inhibitor, 100 nM), pazopanib (VEGFR inhibitor, 100 nM), UNC2025 (MERTK inhibitor, 300 nM), UNC4241 (MERTK inhibitor, 300 nM), BKM120 (PI3Kα inhibitor, 100 nM), IPI549 (PI3Kγ inhibitor, 100 nM), and Torin 2 (mTOR inhibitor, 100 nM). **b** IB analysis of WCL derived from RCC4 cells deleted of endogenous MERTK using indicated sgRNAs by CRISPR-Cas9. **c** IB analysis of WCL derived from UMRC6 cells depleted of endogenous MERTK using indicated shRNAs. **d** Deletion of *MERTK* in RCC4 cells leads to reduced colony formation ability. **e**, **f** IB analysis of WCL derived from RCC4 (**e**) or UMRC6 (**f**) cells treated with indicated MERTK inhibitors with indicated doses for 2 h. **g** In vitro kinase assays to demonstrate that MERTK directly phosphorylates the Akt1-Y26 in vitro. **h–j** IB analysis of WCL derived from RCC4 cells treated with 80 μM phosphatidylserine (PtdSer) (**h**) or 10 μM pervanadate (PV) (**i**, **j**) for 30 min in the absence or presence of 300 nM MERTK inhibitor UNC2025 (**h**, **i**) or MERTK deletion by CRISPR (**j**). **k**, **l** IB analysis of WCL derived from RCC4 cells serum starved overnight and then treated with indicated stimulation and inhibitors. EGF (100 ng/ml, 10 min), insulin (100 nM, 30 min), PV (60 μM, 30 min), BKM120 (200 nM, 10 h), IPI549 (100 nM, 10 h), UNC2025 (300 nM, 2 h). **m** IB analysis of WCL and PI(3,4,5)P₃ beads pulldown products derived from RCC4 cells transfected with indicated DNA constructs. **n** IB analysis of WCL derived from *DLD1-Akt1/2⁻/⁻* cells transfected with indicated HA-Akt1 constructs. **o** A cartoon illustration indicating that MERTK-mediated Akt1-Y26 phosphorylation releases SAV1 binding and promotes Akt activation

(Supplementary Fig. 3m), further supporting the notion that SAV1 is an Akt activity suppressor. Together, these data support that SAV1 binds and suppresses Akt activation independent of its canonical SARAH domain-dependent function in Hippo/YAP regulation (Supplementary Fig. 3n).

Moreover, ectopic expression of SAV1 in 786-O cells ($SAV1^{-/-}$) significantly reduced Akt-T308 phosphorylation in response to EGF (Supplementary Fig. 4a), while depletion of endogenous SAV1 facilitated Akt-T308 phosphorylation upon EGF (Supplementary Fig. 4b and 4c) or insulin (Supplementary Fig. 4d) stimulation. These data further support that SAV1 suppresses Akt activation under growth factor stimulation conditions.

**MERTK phosphorylates Akt to release SAV1 binding.** Given that phosphorylation of Akt1-Y26 disrupts SAV1 binding (Fig. 1f), we sought to identify tyrosine kinase(s) that phosphorylates Akt1-Y26, which would promote SAV1 release allowing for Akt activation. We reasoned that suppressing the activity of target kinase(s) responsible for Akt1-Y26 phosphorylation would result in enhanced Akt1 interaction with SAV1, mimicking the behavior of the Akt1-Y26F mutation (Fig. 1e). We tested a series of tyrosine kinase inhibitors that target commonly hyper-activated tyrosine kinases in cancer. Interestingly, the broad-spectrum inhibitor dasatinib did not alter the interaction, but MERTK inhibition by two MERTK selective inhibitors (UNC2025 and UNC4241) significantly promoted SAV1 binding to Akt1 (Fig. 2a and Supplementary Fig. 5a and 5b), as well as suppressed Akt-pT308 (Supplementary Fig. 5c) in a Y26 phosphorylation-dependent manner (Supplementary Fig. 5d). Genetic ablation of *MERTK* by CRISPR (Fig. 2b), or depletion of MERTK by shRNAs (Fig. 2c and Supplementary Fig. 5e), led to significantly reduced Akt-pT308 signals in RCC4 cells and subsequently attenuated RCC4 cell growth (Fig. 2d). The association with MERTK was further supported by reduced Akt-T308 phosphorylation in $mertk^{-/-}$ mouse kidney tissues; the reduction was larger compared with kidney tissues isolated from either $tyro3^{-/-}$ or $axl^{-/-}$ mice (Supplementary Fig. 5f and 5g). In addition, CRISPR-mediated deletion of *Axl* or *Tyro3*, the two other Tyro3/Axl/MerTK (TAM) kinase family members, did not result in significantly reduced Akt-T308 phosphorylation as compared to MERTK (Fig. 2b, c, and Supplementary Fig. 5h to 5k). Furthermore, deletion of *Axl* did not retard cell growth (Supplementary Fig. 5l and 5m). Moreover, acute MERTK inhibition by UNC2025 significantly reduced Akt-pT308 in both RCC4 (Fig. 2e) and UMRC6 cells (Fig. 2f). In addition, active MERTK could directly phosphorylate Akt1 in vitro mainly on the Y26 residue (Fig. 2g), further supporting MERTK as a major kinase responsible for Akt1-Y26 phosphorylation. Consistent with this notion, a moderate correlation between Akt-pT308 and MERTK activity (evidenced by MEK1-pS217/pS221[32]) was observed in Stage IV KIRC patients TCGA dataset (Supplementary Fig. 5n to 5q).

More importantly, phosphatidylserine (PtdSer), the complex physiological activator for MERTK[33], triggered Akt-pT308 in a MERTK kinase activity-dependent manner (Fig. 2h and Supplementary Fig. 6a), in the presence of GAS6 (Supplementary Fig. 6b and 6c) produced by RCC4 cells[34]. Moreover, GAS6 also promoted Akt1-pY26 and Akt-pT308 in RCC4 cells (Supplementary Fig. 6d). In keeping with the notion that physiological activation of MERTK promotes Akt activation, we also observed that expression of a constitutively-active MERTK, Fc-MERTK[35], greatly facilitated Akt-T308 phosphorylation in response to either EGF (Supplementary Fig. 6e) or insulin (Supplementary Fig. 6f) stimulation. On the other hand, *MERTK* deletion in RCC4 cells resulted in attenuated Akt-T308 phosphorylation in response to either EGF (Supplementary Fig. 6g) or insulin

stimulation (Supplementary Fig. 6h). These data cumulatively suggest MERTK as a major kinase phosphorylating Akt1-Y26 for Akt activation. Thus, in the remainder of the study, we mainly focus on understanding the role of MERTK in activating Akt; it remains to be determined whether other RTKs or intracellular TKs also play a role in this process. However, it is interesting that the broad src family TK inhibitor, dasatinib, does not significantly affect SAV1 binding to Akt (Fig. 2a).

To further explore the connection between MERTK phosphorylation of Akt1-Y26 and to explore the physiological function of Akt1-Y26 phosphorylation in cells, we generated an Akt1-pY26-specific antibody (Supplementary Fig. 7a and 7b). Pervanadate (PV), a tyrosine phosphatase inhibitor leading to MERTK activation[32], promoted Akt-Y26 phosphorylation and Akt-T308 phosphorylation (Supplementary Fig. 8a to 8c) in a MERTK-dependent manner (Fig. 2i, j and Supplementary Fig. 8d and 8e). More importantly, compared with growth factor stimulation by EGF or insulin, PV treatment led to much more significant Akt1-Y26 phosphorylation (Fig. 2k), which could be specifically suppressed by inhibitors targeting MERTK (by UNC2025), but not PI3Kα (by BKM120) nor PI3Kγ (by IPI549) (Fig. 2l and Supplementary Fig. 8f). EGFR and insulin receptors are well-known activators of canonical Akt pathways, but confirming their minimal involvement in this mechanism, neither EGF (Supplementary Fig. 9a) nor insulin (Supplementary Fig. 9b) treatment significantly influenced Akt interaction with SAV1. On the other hand, MERTK activation (by Fc-MERTK expression)-induced Akt1-pY26 increase led to attenuated SAV1 binding (Supplementary Fig. 9c).

To further investigate the contribution of the MERTK/Akt/SAV1 signaling in Akt activation triggered by growth signaling such as PI3K, we ectopically expressed a cancer patient-derived constitutively active H1047R-PIK3CA mutant in RCC cells depleted of endogenous SAV1. Although ectopic expression of either H1047R-PIK3CA or EGFR-L858R promoted basal Akt-T308 phosphorylation (Supplementary Fig. 10a to 10c), in response to growth stimulations including EGF (Supplementary Fig. 10d and 10e) and insulin (Supplementary Fig. 10f), hyperactivation of PI3K did not significantly promote Akt-pT308 signals in SAV1-depleted RCC cells. Similarly, expression of H1047R-PIK3CA also did not functionally compensate for loss of *MERTK* in promoting phosphorylation of Akt-T308 upon EGF (Supplementary Fig. 10g) or insulin (Supplementary Fig. 10h) stimulation. Moreover, expression of a constitutively active EGFR-L858R mutant similarly did not dramatically promote Akt-pT308 in the absence of neither SAV1 nor MERTK (Supplementary Fig. 10i and 10j) under stimulation conditions in RCC4 cells. Together, these data suggest that MERTK is a major regulator of Akt1-Y26 phosphorylation and subsequent Akt activation, a physiological/potentially pathophysiologic stimulatory pathway distinct from canonical growth factor-mediated[6] Akt activation mechanisms.

Next, we examined the molecular mechanism(s) responsible for SAV1 and MERTK-mediated Akt activity control showing that SAV1 binding to the Akt-PH domain (Supplementary Fig. 1e) restrained its accessibility to PI(3, 4, 5)$P_3$ (Fig. 2m). However, attenuating Akt plasma membrane recruitment was not the only mechanism through which SAV1 suppresses Akt activation. Introduction of the Y26F mutation into a myristoylation-tagged, active Akt1 that is constitutively attached to the plasma membrane still resulted in reduced Akt activity (Fig. 2n), due to enhanced binding to SAV1 (Supplementary Fig. 11a). Furthermore, we found that enhanced SAV1 binding attenuated Akt binding to its upstream activating kinases, including PDK1 that phosphorylates Akt-T308[12] (Supplementary Fig. 11b to 11d) and Sin1 (Supplementary Fig. 11e), an essential mTORC2 component for Akt-S473 phosphorylation[11] at plasma

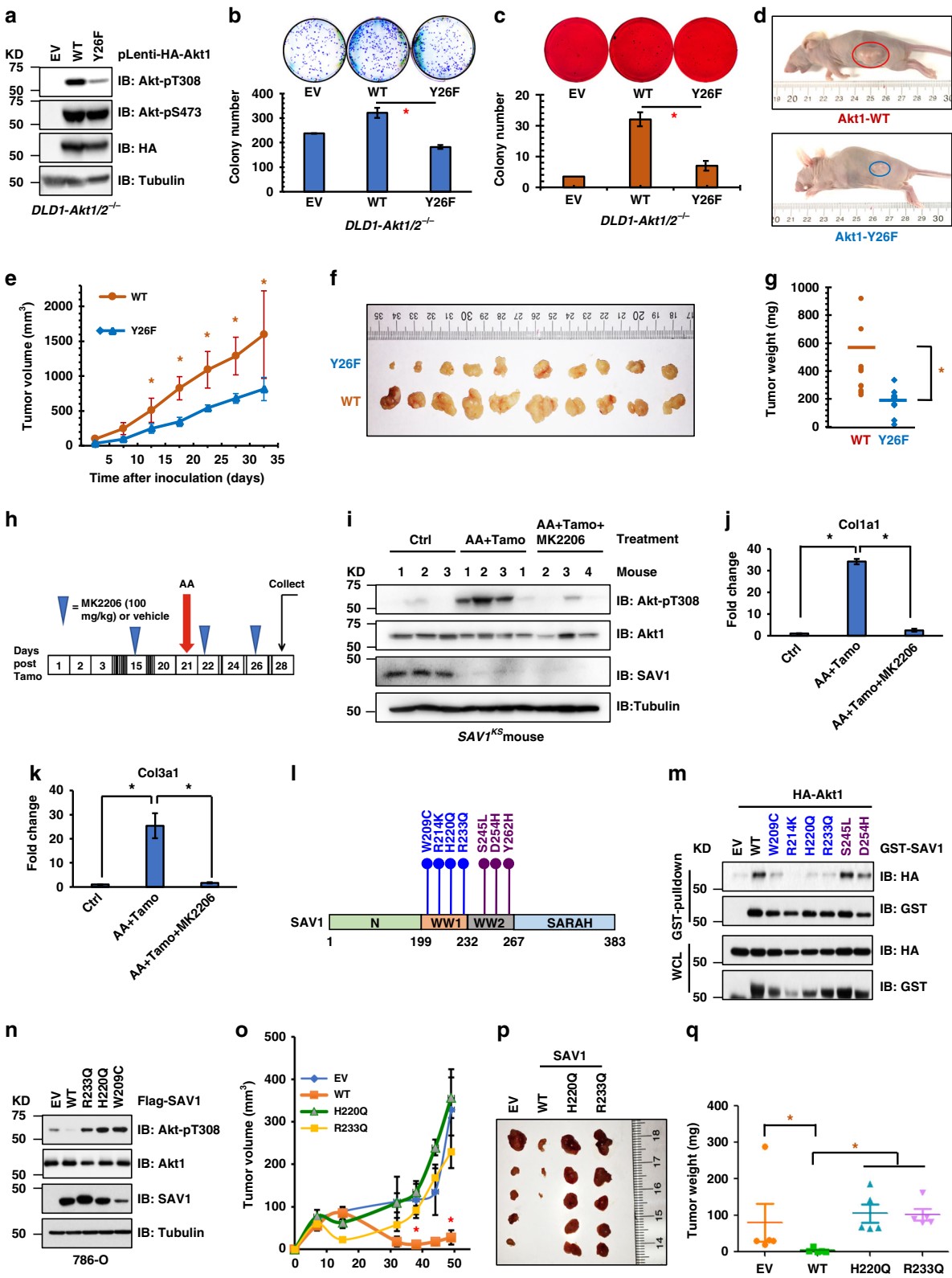

membrane[36]. Thus, SAV1 suppresses Akt activation even once Akt is recruited to the plasma membrane. Together, these data suggest that SAV1 binding not only diminishes Akt recruitment to the plasma membrane lipids but also interferes with subsequent kinase-mediated Akt phosphorylation and activation processes (Fig. 2o). MERTK-mediated Akt-Y26 phosphorylation releases SAV1 binding and suppression, allowing for Akt plasma membrane attachment and subsequent phosphorylation-mediated activation (Fig. 2o). Together, these data suggest that Akt-pY26 is a control mechanism for Akt activation in SAV1-expressing cells.

**Akt1-Y26 phosphorylation contributes to Akt oncogenicity.** Next, we examined whether enhanced SAV1 binding to Akt, or

**Fig. 3** Akt1-Y26 phosphorylation contributes to the Akt oncogenic capacity. **a** IB analysis of WCL derived from *DLD1-Akt1/2$^{-/-}$* cells infected with lentiviruses encoding indicated Akt1 constructs and selected with 1 μg/ml puromycin for 72 h before cell collection. **b, c** Colony formation (**b**) and soft agar (**c**) assays were performed with Akt1-WT or Akt1-Y26F expressing DLD1-*AKT1/2$^{-/-}$* cells generated in (**a**), and were quantified (mean ± SD, n = 3). *P < 0.05 (Student's t test). **d–g** Mouse xenograft experiments were performed with the cells generated in (**a**). Representative tumor images (**d**), tumor growth curve (**e**), and tumor weight (**f, g**) were calculated (mean ± SD, n = 10). *P < 0.05 (one-way ANOVA test). **h** A schematic representation for drug treatment on *SAV1$^{KS}$* mice as indicated. **i** IB analysis of WCL derived from *SAV1$^{KS}$* mouse kidneys receiving indicated treatments. **j, k** Analyses of mRNA changes in Col1a1 and Col3a1 fibrosis genes in *SAV1$^{KS}$* mouse kidneys receiving indicated treatments. **l** A schematic representation of cancer patient-associated SAV1 WW-domain mutations. **m** IB analysis of WCL and GST-pulldowns derived from HEK293 cells transfected with indicated SAV1-constructs with HA-Akt1. **n** IB analysis of WCL derived from 786-O cells (*SAV1$^{-/-}$*) transfected with indicated SAV1 constructs. **o–q** Mouse xenograft experiments were performed with the indicated 786-O cells. Representative tumor growth curve (**o**), tumor images (**p**), and tumor weight (**q**) were calculated (mean ± SD, n = 10). *P < 0.05 (one-way ANOVA test)

deficiency in MERTK-mediated Akt1-Y26 phosphorylation regulates Akt activity- governed cell growth. We reconstituted *DLD1-Akt1/2$^{-/-}$* cells with a stable expression of either WT- or Y26F-Akt1 (Fig. 3a). Consistent with our previous observations, Akt1-Y26F expressing cells displayed significantly reduced Akt-pT308 signals (Fig. 3a and Supplementary Fig. 12a and 12b), and subsequently attenuated colony formation (Fig. 3b). Moreover, anchorage-independent growth (Fig. 3c) and tumor formation in a mouse xenograft model (Fig. 3d–g) were also reduced significantly. These data indicate that SAV1 binding to Akt may limit tumorigenesis. Given that we previously found that conditional knockout of *Sav1* in mouse renal tubule cells results in renal fibrosis upon acute kidney injury (AKI) with aristolochic acid (AA), concomitant with an oncogene-induced senescence phenotype[25], next we went on to examine whether it is in part due to *Sav1*-deletion induced Akt hyperactivation. Consistent with our previous observation, AA treatment in *Sav1$^{KS}$* mice induced expression of genes associated with fibrosis (Supplementary Fig. 12c). More importantly, inhibition of Akt by MK2206 (Fig. 3h, i) significantly suppressed the expression of these fibrosis genes including Col1a1 (Fig. 3j) and Col3a1 (Fig. 3k). These data strongly support that Akt activation contributes to the genetic and phenotypic manifestations of renal fibrosis induced by AA in *Sav1$^{KS}$* mice and that Akt activation may serve as a major signal downstream of SAV1 deficiency.

Considering that key cell signaling events could be hijacked by cancer to facilitate tumorigenesis[37], we analyzed TCGA datasets, identified and generated a set of mutations in the WW domains of SAV1 (Fig. 3l). Mutations in the functional WW1 domain (W209C, R214K, H220Q, and R233Q), but not in the non-functional WW2 domain (S245L and D254H), displayed significantly reduced binding to Akt1 (Fig. 3m) and elevated Akt-pT308 signals (Fig. 3n). Introducing WT-SAV1, but not the H220Q nor R233Q WW1 domain mutant, into 786-O cells (which lack SAV1 expression) suppressed 786-O cell growth both in nude mice (Fig. 3o–q, Supplementary Fig. 12d and 12e) and in vitro (Supplementary Fig. 12f and 12g). These data strongly suggest that the SAV1-WW1 domain mutations facilitate tumorigenesis potentially through their inability to suppress Akt activation.

**Akt suppression by MERTK inhibition requires SAV1 in RCC.** Given that SAV1 binds and suppresses Akt through its PH domain (Supplementary Fig. 1e), and allosteric Akt inhibitors specifically target the Akt-PH domain to inhibit Akt plasma membrane attachment and activation[38], we examined whether SAV1 could antagonize the ability of Akt allosteric inhibitors to suppress Akt activity. We found that the allosteric Akt inhibitor, MK2206 (Fig. 4a), but not an ATP analogous inhibitor, GDC0068 (Fig. 4b), specifically competed with SAV1 to bind Akt in an Akt-PH domain-dependent manner (Supplementary Fig. 13a and 13b). Further, SAV1 depletion led to increased cellular sensitivity

to MK2206 (Fig. 4c); this was not as evident with GDC0068 (Fig. 4d). These data suggest that reduced SAV1 may sensitize cells to Akt allosteric inhibitors resulting from increased accessibility of Akt-PH domains for these inhibitors.

MERTK plays physiologic roles in innate immune cells and epithelium (including renal epithelia cells[39]) sensing PtdSer on apoptotic, ER stressed cells, exosomes, and platelets. It is overexpressed in a variety of neoplastic cells where it provides a survival signal, in part related to Akt activation that has been demonstrated by multiple studies[40]. Examination of the MERTK control of this SAV1 mechanism revealed that when endogenous SAV1 was depleted, MERTK inhibition, which suppressed Akt activation (Fig. 2e, f) now failed to efficiently suppress Akt activation (Fig. 4e and Supplementary Fig. 13c), and these cells were more resistant to MERTK inhibition-mediated growth suppression (Fig. 4f, g), suggesting that SAV1 may be a major downstream effector of MERTK survival signaling in neoplasia and may be a marker to assess whether MERTK inhibition will be effective. Similarly, MERTK inhibition could not further suppress Akt activation when Akt1-Y26 phosphorylation was deficient (Supplementary Fig. 13d), suggesting that MERTK inhibition-mediated Akt suppression is largely through Akt1-Y26 phosphorylation controlled SAV1 loading. Consistent with this notion, unlike WT-SAV1, the cancer-associated SAV1-R233Q mutant was largely resistant to MERTK inhibitor (UNC2025)-mediated suppression of Akt activation (Fig. 4h) and displayed less sensitivity to UNC2025 (Fig. 4i).

Together, these data collectively suggest that inhibition of MERTK reduces phosphorylation of Akt1-Y26, which increases SAV1-Akt association (Fig. 4j). This model suggests that cancers lacking SAV1 may have a limited response to MERTK inhibitors, a connection that warrants additional investigations.

## Discussion

Our studies reveal that in spite of intense study of the regulation of a crucial oncogenic driver, Akt, there is still much to learn. Reasoning that cell growth and cell size control might be related we found that SAV1 operates in both pathways with SAV1 being an endogenous Akt inhibitor through its WW domain interaction with a PxY motif in the Akt PH domain. The interaction provides at least two levels of suppression: restriction of PH binding to the plasma membrane and restriction of interaction with the Akt membrane activating kinases, PDK1 and mTORC2. This additional level of control can be abrogated by tyrosine phosphorylation of the PxY motif, which is much less effectively accomplished by two major kinase systems usually associated with Akt control including EGFR and insulin receptor/IGFR1[6]. Instead, we demonstrate that MERTK is one RTK that has properties allowing it to regulate this mechanism. Whether other RTKs or TKs can regulate Y26 phosphorylation remains to be determined, but our data suggest that the other two members of the PtdSer sensing RTKs, Axl and Tyro3, are not as efficient in

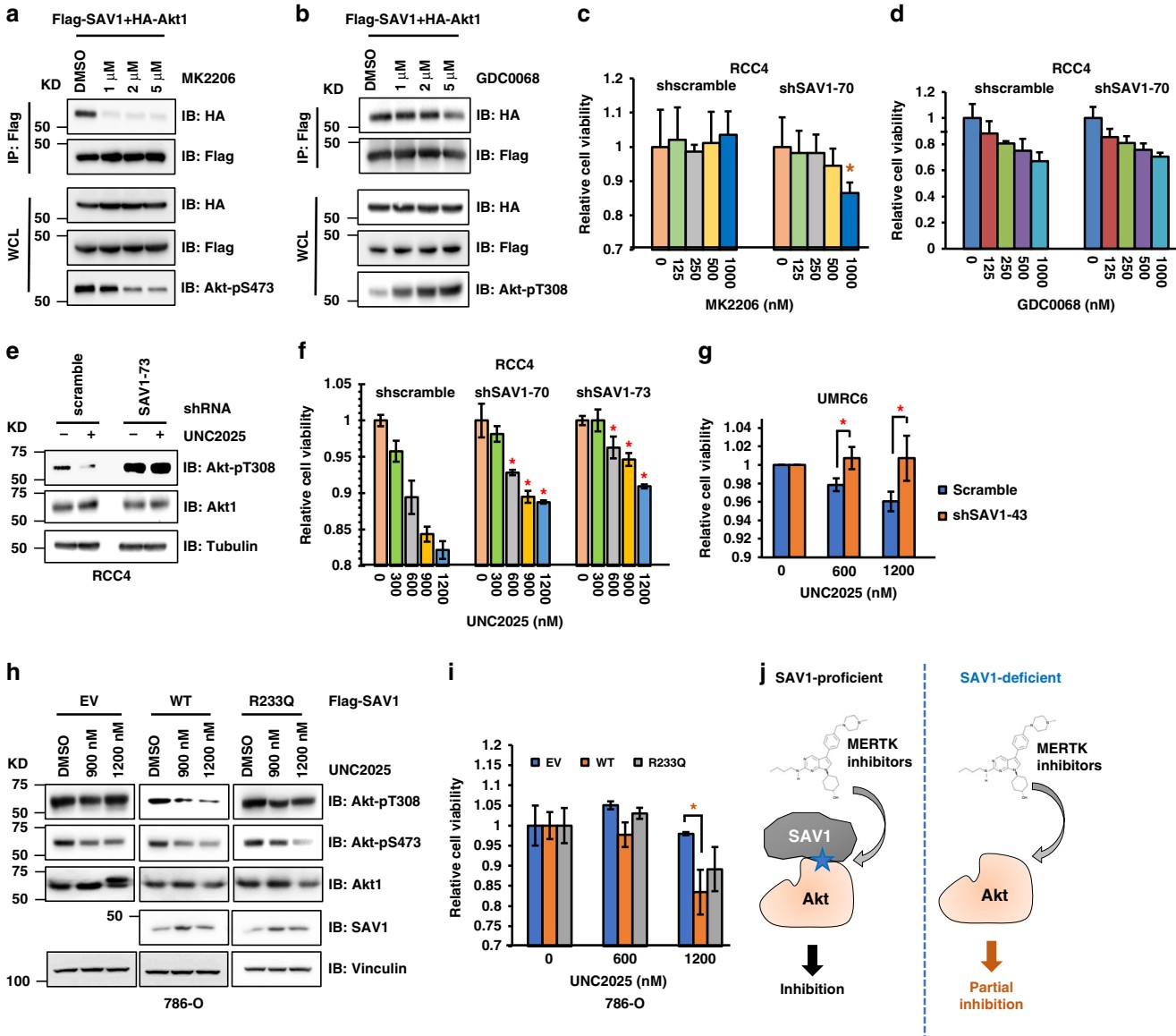

**Fig. 4** SAV1 is necessary for MERTK inhibition-mediated suppression of Akt activation in RCC cells. **a**, **b** IB analysis of WCL derived from HEK293 cells transfected with Flag-SAV1 and HA-Akt1. Where indicated, cells were treated with indicated doses of MK2206 (**a**) or GDC0068 (**b**) for 10 h before cell collection. **c**, **d** Cell viability assays were performed with WT- and SAV1-depleted RCC4 cells treated with either indicated doses of MK2206 (**c**) or GDC0068 (**d**) for 3 days (mean ± SD, n = 3). 1000 cells were plated in each well on 96-well plates. *P < 0.05 (Student's t test). **e** IB analysis of WCL derived from indicated RCC4 cells treated with MERTK inhibitor UNC2025 (400 nM) for 2 h before cell collection. **f**, **g** Cell viability assays were performed with WT- and SAV1-depleted RCC4 (**f**) or UMRC6 (**g**) cells treated with indicated doses of MERTK inhibitor UNC2025 for 3 days (mean ± SD, n = 3). *P < 0.05 (Student's t test). **h** IB analysis of WCL derived from 786-O cells transfected with indicated SAV1 constructs and treated with indicated doses of MERTK inhibitor UNC2025 (400 nM) for 2 h before cell collection. **i** Cell viability assays were performed with 786-O cells transfected with indicated SAV1 constructs and treated with indicated doses of MERTK inhibitor UNC2025 for 3 days (mean ± SD, n = 3). *P < 0.05 (Student's t test). **j** A schematic proposed model to demonstrate that SAV1 is necessary for MERTK-inhibition governed inactivation of Akt. Specifically, when SAV1 expression is high, MERTK inhibitors reduce Akt1-pY26 signals, subsequently enhancing SAV1 binding to Akt to block Akt plasma membrane attachment and binding to Akt upstream activating kinases, thus leading to suppression of Akt. On the other hand, when SAV1 expression is low, MERTK inhibition, although still reduces Akt1-pY26 signals, would achieve limited effects due to the inability to recruit SAV1 for suppressing Akt

carrying out this regulation of a crucial oncogenic driver. Perhaps the larger size of MERTK (170 kDa) compared to Axl (~138 kDa) and Tyro3 (~130 kDa) gives MERTK a structural/cellular advantage. In epithelial tumors, most of which express MERTK at some level and all of which have abundant apoptotic materials stimulating MERTK activation, this mechanism phosphorylating Akt-Y26, releasing SAV1 (in WT-SAV1 expressing cells), allowing Akt movement to the plasma membrane and unfettered access to PDK1 and mTORC2 would promote oncogenic survival

signaling. The biochemical/stoichiometric relationship between the canonical drivers of PI-3K and this mechanism remains to be explored. Notably, cancer therapy-induced cellular apoptosis will lead to accumulation of cellular apoptotic materials including PtdSer, Gas6, and ProteinS, which may contribute to survival advantages essential for therapy resistance. Consistent with this notion, recently activation of the Gas6/TAM signaling upon cytarabine treatment is indispensable for AML resistance to cytarabine[41]. It remains to be further determined whether TAM

ligands-mediated TAM activation is a general mechanism contributing to cancer therapy-induced therapeutic resistance by promoting Akt-Y26 phosphorylation and Akt activation.

Tumorigenic Akt activation now has other potential mechanisms including deficiency in SAV1 expression, SAV1 mutations in WW domains, which are known to occur[26], or hyperactivation of MERTK[35]. SAV1 appears to be an important component for MERTK-inhibition triggered Akt inactivation, and the SAV1 expression status could be one marker of whether MERTK inhibitors, which are entering clinical trials, will be efficacious as anti-tumor cell-directed therapies. Our data also suggest that MERTK inhibitors and allosteric Akt inhibitors will have overlapping sites of action. The Akt pathways are funnels through which almost all cancers pass and as such are major therapeutic targets, the present results implicate MERTK as another potential target to decrease Akt activation. The current study reveals that there are still layers of regulation to be uncovered; starting with a hypothesis that the Akt and Hippo signaling may interact, our studies reveal an additional mechanism and a path by which a less studied RTK, MERTK, may regulate this mechanism. These results could provide a therapeutic target and perhaps a predictive marker to guide clinical trials with MERTK and potentially other TKs that regulate Akt1-Y26 phosphorylation and SAV1/Akt-PH interaction.

## Methods

**Cell culture and transfection.** HEK293, HEK293T, HeLa, T98G, renal carcinoma cell lines including SAV1-deficient cells (ACHN and 786-O), and SAV1-proficient cells (RCC4 and UMRC6) were cultured in DMEM medium supplemented with 10% FBS, 100 units of penicillin and 100 mg/ml streptomycin. HeLa cells were as described in ref. [13]. HEK293T and other RCC cell lines were as described in ref. [42]. DLD1-Akt1$^{-/-}$Akt2$^{-/-}$ cells were kindly provided by Dr. Bert Vogelstein (Johns Hopkins University School of Medicine) and these cells also were maintained in DMEM medium supplemented with 10% FBS. Cell transfection was performed using lipofectamine 2000 or polyethylenimine (PEI) according to manufacture instructions[13,36]. The packaging of lentiviral shRNA or cDNA expressing viruses and retroviral cDNA expressing viruses, as well as subsequent infection of various cell lines were performed according to the adapted protocols[14,43]. Briefly, viral DNA was co-transfected with packaging plasmids including VSVG and Δ8.9 into HEK293T cells and virus-containing media were collected in 2 consecutive days and used for infection of indicated cells. Following viral infection, cells were maintained in the presence of hygromycin (200 μg/ml), puromycin (1 μg/ml), or blasticidin (5 μg/ml), depending on the viral vector used to infect cells.

Akt kinase inhibitors MK2206 (Selleck S1078) and GDC0068 (Selleck S2808), PI3Kα inhibitor BKM120 (Selleck S2247), mTOR inhibitor Torin 2 (Selleck S2817), PI3Kγ inhibitor IPI549 (Selleck S8330), c-Met inhibitor cabozantinib (Selleck S1119), Syc inhibitor dasatinib (Selleck S1021), VEGFR inhibitor pazopanib (Selleck S3012), MERTK inhibitors UNC2025 and UNC4241 (produced by UNC drug discovery unit) were used at the indicated dose(s). EGF (Sigma E9644), insulin (Invitrogen 41400-045), pervanadate (Sigma cat#069K0031), and phosphatidylserine (Avanti, Polar Lipids cat#840032P) were used at the indicated doses. PIP3 beads (P-B00Ss) were purchased from Echelon Biosciences. Akt1 peptides were synthesized by LifeTin. EGF (1150-04) was purchased from GoldBio. GAS6 was purchased from R&D Systems (986-GS) or generated by the Earp lab.

**Plasmids construction.** pcDNA3-HA-Akt1, pcDNA3-HA-Akt2, pcDNA3-HA-Akt3, pCMV-Flag-PDK1, and pCMV-Flag-Sin1 were previously described[13]. pCMV-Flag-SAV1, pCMV-Flag-TAZ, pCMV-Flag-YAP1, pCMV-Flag-AYP2 were obtained from Dr. Kun-Liang Guan (UC-San Diego). Flag-PIK3CA-WT and Flag-PIK3CA-H1047R plasmids were kind gifts from Dr. Alex Toker (Beth Israel Deaconess Medical Center, Harvard Medical School). Fc-MERTK plasmid was obtained from Dr. Raymond Birge (Rutgers University). EGFR-WT and EGFR-L858R plasmids were generated in Dr. H. Shelton Earp lab. pLenti-HA-Akt1-WT and Y26F were constructed by cloning corresponding PCR fragments into pLenti-puro-HA vector described previously[13] by BamHI and SalI sites. Plenti-HA-SAV1-WT and various mutants were cloned into pLenti6-HA vector by BamHI and XhoI sites. pCMV-GST-SAV1-full length and truncation constructs, pCMV-GST-Akt1-full length and truncation constructions were cloned into mammalian expression CMV-GST-fusion vectors. Various Akt1 and SAV1 mutants were generated using the QuikChange XL Site-Directed Mutagenesis Kit (Stratagene) according to the manufacturer's instructions. All mutants were generated using mutagenesis PCR and the sequences verified by sequencing. Details of plasmid constructions are available upon request.

Akt1-Y26F-Forward: 5′-CATCAAGACCTGGCGGCCACGCTTCTTCCTCCT CAAGAATG-3′

Akt1-Y26F-Reverse: 5′-CATTCTTGAGGAGGAAGAAGCGTGGCCGCCAGG TCTTGATG-3′

Akt1-P24A-Forward: 5′-CATCAAGACCTGGCGGGCACGCTACTTCCTCCT CAAGAATG-3′

Akt1-P24A-Reverse: 5′-CATTCTTGAGGAGGAAGTAGCGTGCCCGCCAGG TCTTGATG-3′

Akt1-BamHI-Forward: 5′-GCATGGATCCAGCGACGTGGCTATTGTG-3′

Akt1-SalI-Reverse: 5′-GCATGTCGACTCAGGCCGTGCCGCTGGC-3′

**shRNAs and sgRNAs.** shRNA vectors to deplete endogenous SAV1 were purchased from Sigma (SHCLNG-NM_021818). shRNA vectors to deplete endogenous MERTK were purchased from Sigma (SHCLNG-NM_006343). The sequence for sgRNAs for CRISPR-Cas9 mediated deletion of MERTK were generated by cloning the annealed sgRNAs into BsmBI-digested pLenti-CRISPRv2 vector (Addgene 52961). The short guide RNAs (sgRNA) were designed based on predictions from crispr.mit.edu and oligonucleotides were obtained from Eton Biosciences. Lentiviruses were produced by PEI transfection of $8 \times 10^6$ HEK293T cells per 10 cm plate with 2.5 μg of target plasmid combined with 1.25 μg of VSVG and 1.25 μg of Δ8.9 for transfection. CRISPR sgRNAs were designed as listed below:

sgControl-Forward: 5′-CACCGCTTGTTGCGTATACGAGACT-3′;
sgControl-Reverse: 5′-AAACAGTCTCGTATACGCAACAAG-3′;
MERTK-sg1-Forward: 5′-CACCGGTAATTTCTCTCCGGACGGA-3′;
MERTK-sg1-Reverse: 5′-AAACTCCGTCCGGAGAGAAATTACC-3′;
MERTK-sg2-Forward: 5′-CACCGCCCGGGAATAGCGGGTAAGG-3′;
MERTK-sg2-Reverse: 5′-AAACCCTTACCCGCTATTCCCGGGC-3′;
Axl-sg1-Forward: 5′-CACCG CTGCGAAGCCCATAACGCCA-3′;
Axl-Sg1-Reverse: 5′-AAAC TGGCGTTATGGGCTTCGCAG C-3′;
Axl-sg2-Forward: 5′-CACCG CGGGCACCTGTGATATTCCC-3′;
Axl-Sg2-Reverse: 5′-AAAC GGGAATATCACAGGTGCCCG C-3′;
Axl-sg3-Forward: 5′-CACCG GAGAGCCCCCCGAGGTACAT-3′;
Axl-Sg3-Reverse: 5′-AAAC ATGTACCTCGGGGGGCTCTC C-3′;
Axl-sg4-Forward: 5′-CACCG CAGAGCCCGTGGACCTACTC-3′;
Axl-Sg4-Reverse: 5′-AAAC GAGTAGGTCCACGGGCTCTG C-3′;
Tyro3-sg1-Forward: 5′-CACCG GCTCTGACGCCGGCCGGTAC-3′;
Tyro3-sg1-Reverse: 5′-AAAC GTACCGGCCGGCGTCAGAGC C-3′;
Tyro3-sg3-Forward: 5′-CACCG CCCTTTCCAACTGTCTTGTG-3′;
Tyro3-sg3-Reverse: 5′-AAAC CACAAGACAGTTGGAAAGGG C-3′;
Tyro3-sg5-Forward: 5′-CACCG TGTGAAGCTCACAACCTAAA-3′;
Tyro3-sg5-Reverse: 5′-AAAC TTTAGGTTGTGAGCTTCACA C-3′

**Antibodies.** All antibodies were used at a 1:1000 dilution in TBST buffer with 5% non-fat milk for western blotting if not otherwise indicated. Anti-SAV1 antibody (cat13301, lot#1), anti-phospho-Thr308-Akt antibody (cat2965, lot#18), anti-phospho-Ser473-Akt antibody (cat4060, lot#19 and 23), anti-Akt1 antibody (cat#2938, lot#4; and cat#2967, lot#17), anti-phospho-Ser9-GSK3b antibody (cat5558, lot#6), anti-PTEN antibody (cat#9559, lot#17), anti-phospho-Thr389-S6K antibody (cat9234, lot#11), anti-S6K1 antibody (cat#2708), anti-GST antibody (cat#2625, lot#7), anti-phospho42/44-ERK (cat#4370, lot#17), anti-phospho-Thr180/Tyr182-p38 (cat#9211, lot#21), anti-Axl antibody (cat#8661s, lot#4), anti-Phospho-Y702-Axl (cat#5724s, lot#1), anti-Tyro3 antibody (cat#5585, lot#2), and polyclonal anti-phospho-Tyr1086-EGFR antibody (cat#2220s, lot#2) were purchased from Cell Signaling Technology. Polyclonal anti-HA antibody (cat#sc-805, lot#K1215) was obtained from Santa Cruz. Polyclonal anti-Flag antibody (cat#F-7425, lot#078M4886V), monoclonal anti-Flag antibody (cat#F-3165, clone M2, lot#SLBN8915V), monoclonal anti-Tubulin antibody (cat#T-5168, lot#115M4828V), monoclonal anti-Vinculin antibody (cat#V9131), anti-Flag agarose beads (cat#A-2220, lot#SLBW1929), anti-HA agarose beads (cat#A-2095, lot#057M4864V), peroxidase-conjugated anti-mouse secondary antibody (cat#A-4416, lot#SLBW4917), and peroxidase-conjugated anti-rabbit secondary antibody (cat#A-4914, lot#SLBV6850) were obtained from Sigma. Anti-MERTK and anti-EGFR antibodies were generated by the Earp lab.

The polyclonal Akt1-pY26 antibody was generated by Abclonal, Inc. and was derived from rabbits. We have validated this antibody by dot blotting assays with synthetic non-phosphorylated and Y26-phosphorylated Akt1 peptides, as well as in immunoprecipitations and whole cell lysates from Akt1-WT and Akt1-Y26F constructs transfected into HEK293 cells.

**Immunoblot and immunoprecipitations analyses.** Cells were lysed in EBC buffer (50 mM Tris, pH 7.5, 120 mM NaCl, 0.5% NP-40) or Triton X-100 buffer (50 mM Tris, pH 7.5, 150 mM NaCl, 1% Triton X-100) supplemented with protease inhibitors (Complete Mini, Roche) and phosphatase inhibitors (phosphatase inhibitor cocktail set I and II, Calbiochem). The protein concentrations of whole cell lysates were measured by NanoDrop OneC using the Bio-Rad protein assay reagent according to manufacture instructions[13]. Equal amounts of whole cell lysates were resolved by SDS-PAGE and immunoblotted with indicated antibodies. For immunoprecipitations analysis, 1000 μg lysates were incubated with the indicated antibody (1–2 μg) for 3–4 h at 4 °C followed by 1 h incubation with 10 μl Protein A magnetic beads (New England Biolabs). Or 1000 μg lysates containing tagged molecules were incubated with agarose beads coupled antibodies for the specific tag for 3–4 h at 4 °C. The recovered immuno-complexes were washed five times with NETN buffer (20 mM Tris, pH 8.0, 100 mM NaCl, 1 mM EDTA and 0.5% NP-40)

before being resolved by SDS-PAGE and immunoblotted with indicated antibodies. Uncropped images are provided in Supplementary Fig. 14.

**In vitro Akt kinase assays**. Akt in vitro kinase assays were adapted from a protocol described previously[13]. Briefly, 2 µg of immune-precipitated HA-Akt1-WT or HA-Akt1-Y26F from HEK293 cell lysates were incubated in the presence of 1 µCi-$^{32}$P-ATP and 200 µM cold ATP in the kinase reaction buffer (New England Biolabs B6022) for 30 min at 30 °C. The reaction was stopped by the addition of SDS-containing buffer and resolved by SDS-PAGE. Akt auto-phosphorylation was detected by autoradiography.

**In vitro MERTK kinase assays**. MERTK in vitro kinase assays were adapted from a protocol[13]. Briefly, 2 µg of purified GST-Akt1-WT-K179M or GST-Akt1-Y26F-K179M from HEK293 cells were incubated with immunoprecipitated MERTK by MERTK antibodies from RCC4 cells in the presence of 1 µCi-$^{32}$P-ATP and 200 µM cold ATP in the kinase reaction buffer (New England Biolabs B6022) for 30 min at 30 °C. The reaction was stopped by the addition of SDS-containing buffer and resolved by SDS-PAGE. MERTK-mediated Akt phosphorylation was detected by autoradiography.

**Peptide synthesis**. All peptides were synthesized at LifeTin. Each contained an N-terminal biotin and free C-terminus and was synthesized in the 1–4 mg scale. The sequences are listed below:
Biotin-Akt1-WT: Biotin-GEYIKTWRPRYFLLKNDGTF
Biotin-Akt1-Y26p: Biotin-GEYIKTWRPR-pY-FLLKNDGTF

**Dot immunoblot assays**. Peptides were spotted onto nitrocellulose membrane allowing the solution to penetrate (usually 3–4 mm diameter) by applying it slowly as a volume of 2 µl. The membrane was dried, and blocked in non-specific sites by soaking in TBST buffer with 5% non-fat milk for immunoblot analysis[13].

**Peptide-binding assays**. Indicated biotin-labeled peptides (2 µg) were incubated with 1 mg of whole cell lysates prepared from cells in a total volume of 500 µl for 4 h at 4 °C and 10 µl Streptavidin agarose beads (Thermo Scientific 20353) were added for incubation for another 1 h. The agarose was washed four times with NETN buffer. Bound proteins were eluted by boiling in SDS loading buffer, and resolved by SDS-PAGE for subsequence western blotting.

**Computational modeling**. We used various computational techniques to identify a structural model for Akt1/Sav1 complex. We started our modeling by performing protein–protein docking of the proteins using ClusPro server[44–48]. Protein Data Bank entrees 1H10 [49] and 2YSB (to be published) were used as initial structures of Akt1 and Sav1, respectfully. Based on the proximity to expected binding region three docking models were selected. For further optimization and equilibration of the models, we performed a series of low temperature discrete molecular dynamics (DMD) simulations of the protein complexes. DMD is a computationally efficient approach for molecular dynamics simulation of biological molecules[50,51]. It has enhanced sampling capability compare to traditional molecular dynamics approaches due to efficient even based integration algorithms[52]. For each docking model equilibrium representative was chosen by performing distance base clustering of all structures obtained during 106 time steps of DMD simulations. The centroids of most populated lowest energies clusters are selected as representatives for our equilibrium models.

In order to select one of the models we used our in-house developed tool Eris to compute changes in stability of the complexes due to mutations: Y26F, W209C, R214K, H220Q and compare them to experimental data. Eris[53] is a computationally efficient algorithm that uses a physically based force field with atomic resolution and computationally optimized side chains repacking and backbone relaxation algorithms[54]. This approach allows for an accurate calculation of the effect of one or several mutations on protein stability, ΔΔG. The model that mostly satisfied with the experimental data is presented in Supplementary Fig. 1g. To study the effect of the Akt1-Y26F mutation we analyzed the dynamics of the Akt1 Sav1 complex on 50 ns timescale by performing DMD simulations for both WT and Y26F mutant. We found that the Y26F mutation increases the hydrophobic surface on the interface of the Akt1/SAV1 complex that promotes rearrangement of the binding interface and results in increasing the number of polar interactions stabilizing the Akt1/Y26F-SAV1 complex (Supplementary Fig. 1j and 1k).

**TCGA data analysis**. Scatter plot and Pearson correlation analyses were performed for SAV1 mRNA expression vs. Akt-pT308, Akt-pS473, GSK3-pS9, mTOR, PDK1, and other RPPA using TCGA KIRC data, in Stage I, II, III, IV patient samples, respectively. Similar analyses were performed in TCGA KIRP data (without stage information). In addition, Spearman correlation was also performed for SAV1 mRNA expression vs. Akt-pT308 RPPA, Akt-pS473 RPPA, and others as indicated in Stage IV KIRC, where the data from SAV1 mRNA expression vs. Akt-pT308 RPPA and SAV1 mRNA expression vs. GSK3-pS9 RPPA showed moderate correlations. Moreover, survival analysis was performed for SAV1 mRNA

expression and Akt-pT308 RPPA using TCGA KIRC data. The survival analysis is significant in all the data, but not significant in Stage I, II, III, IV separately.

**RT-PCR**. RT-PCR primers for detection of GAS6 mRNAs are as below:
hGAS6-RT-PCR-F1: 5′-GCAAAACCTGCCTGACCAGT-3′
hGAS6-RT-PCR-R1: 5′-TTCGTTGACATCTTTGTCGCA-3′
β-actin-F: 5′-CCTGGCACCCAGCACAAT-3′
β-actin-R: 5′-GCCGATCCACACGGAGTA-3′
RT-PCR experiments were performed as below:
Total cellular RNA was extracted using RNeasy mini kit (QIAGEN) according to the manufacturer's protocol and quantified by a spectrophotometer (Nanodrop One C). cDNA was synthesized using iScript$^{TM}$ Reverse Transcription Supermix for RT-qPCR (Bio-Rad Cat#1708841). cDNA templates and iTaq$^{TM}$ universal SYBR Green Supermix (Bio-Rad Cat#1725122) were mixed together and the RT-PCR reaction was performed on the ViiA$^{TM}$ 7 Real-Time PCR system. The expression of GAS6 and PROS mRNAs was normalized to the expression of β-actin.

**Colony formation assays**. Indicated cells were seeded into 6-well plates (300 or 600 cells/well) and cultured in 37 °C incubator with 5% $CO_2$ for ~14 days until the formation of visible colonies. Colonies were washed with 1× PBS and fixed with 10% acetic acid/10% methanol for 30 min, stained with 0.4% crystal violet in 20% ethanol for 30 min, and washed by tap water and air-dried. Colony numbers were manually counted. At least two independent experiments were performed to generate the error bars.

**Soft agar assays**. The anchorage-independent cell growth assays were performed according to adapted protocols[13,36]. Briefly, the assays were preformed using 6-well plates where the solid medium consists of two layers. The bottom layer contains 0.8% noble agar and the top layer contains 0.4% agar suspended with $1 \times 10^5$ or $3 \times 10^4$ cells. 500 µl complete DMEM medium with 10% FBS was added every 4 days. About 4 weeks later the cells were stained with iodonitrotetrazolium chloride (1 mg/ml) (Sigma I10406) overnight for colony visualization and counting. At least two independent experiments were performed to generate the error bar.

**Mouse xenograft assays**. Mouse xenograft assays were performed according to adapted protocols[13,36]. Briefly, for mouse xenograft experiments in Fig. 3d–g, $2 \times 10^6$ DLD1-Akt1/2$^{-/-}$ cells stably expressing WT or Y26F mutant form of Akt1 were injected into the flank of 10 female nude mice (NCRNU-M-M from UNC Animal Facility, 4 weeks old). Tumor size was measured every 3 days with a digital caliper, and the tumor volume was determined with the formula: $L \times W^2 \times 0.52$, where $L$ is the longest diameter and $W$ is the shortest diameter. After 18 days, mice were sacrificed, and tumors were dissected and weighed. For mouse xenograft experiments in Fig. 3o–q, $2 \times 10^6$ 786-O cells expressing indicated SAV1 mutant forms were injected into the flank of indicated male nude mice (NCRNU-M-M from UNC Animal Facility, 4 weeks old). Tumor size was measured with a digital caliper at indicated days post-injection, and the tumor volume was determined with the formula: $L \times W^2 \times 0.52$, where $L$ is the longest diameter and $W$ is the shortest diameter. After 49 days, mice were sacrificed, and tumors were dissected and weighed.

**RNA extraction and cDNA synthesis**. Cortex tissue was homogenized in RLT buffer (RNeasy kit; Qiagen) by use of a TissueRuptor homogenizer (Qiagen). RNA was extracted using a RNeasy kit (Qiagen) following the manufacturer's protocol. cDNA was synthesized iScript reverse transcription kit (Bio-Rad).

**Reagents for animal studies**. *Tamoxifen*: Conditional activation of Cre recombinase was induced by oral administration of tamoxifen (Sigma). Tamoxifen (100 mg) was resuspended in 100 µl of ethanol, followed by the addition of 1 ml of sunflower oil, to a final concentration of 100 mg/ml. The solution was sonicated at 4 °C in a water bath sonicator until the tamoxifen was completely dissolved. Animals (8–9 weeks old) were administered 50 µl (5 mg) of tamoxifen solution per day for 3 consecutive days by oral gavage.

*AA*: AA was suspended in 120 L of dimethyl sulfoxide (DMSO) and then 880 µl of PBS (pH 7.4), to a final concentration of 2 mg/ml. The AA solution was administered to mice by intraperitoneal injection at a dosage of 5 mg/kg of body weight. Tissue was extracted for analysis 1 week after AA injection unless stated otherwise.

*MK2206*: MK2206 was resuspended in 1× PBS (pH 7.4), to a final concentration of 100 mg/ml. The MK2206 solution was vortexed vigorously immediately prior to use. Mice were administered MK2206 by oral gavage at a dosage of 100 mg/kg.

All of the necessary animal procedures were performed at the UNC mouse facility, located on the basement of the Genetic Medicine Building (GMB). The mice were housed and maintained under sterile conditions in facilities approved by the American Association for the Accreditation of Laboratory Animal Care and in accordance with current regulations and standards of the United States Department of Agriculture, United States Department of Health and Human Services, and the NIH. The mice were used in accordance with Animal Care and Use Guidelines of The University of North Carolina at Chapel Hill under protocols to Drs. Pengda Liu and William Y. Kim approved by the Institutional Animal Care and Use Committee.

**Statistics**. Differences between control and experimental conditions were evaluated by Student's *t* test or one-way ANOVA. These analyses were performed using the SPSS 11.5 Statistical Software and $P \leq 0.05$ was considered statistically significant.

**Reporting summary**. Further information on experimental design is available in the Nature Research Reporting Summary linked to this article.

## Data availability

All data supporting the findings in this study are available from the corresponding author upon reasonable request.

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

## Acknowledgements

The authors thank Liu, Earp, and Kim lab members for critical reading of the manuscript and helpful discussions. The authors also thank Dr. Qing Zhang (UNC-Chapel Hill) and Zhang lab members for sharing the reagents and suggestions. This work was supported by the NIH grants (P.L. R00CA181342, H.S.E. CA016086, N.V.D. R01GM114015 and R01GM123247), the UNC IBM Junior Faculty Development Award (P.L.), the V Scholar Research Grant (P.L.), and UNC University Cancer Research Fund (P.L.).

## Author contributions

Conceptualization: W.Y.K., H.S.E., and P.L. Methodology: N.V.D. and C.M.P. Investigation: Y.J., Y.Z., J.Y.L., C.F., K.I.P., S.S., J.Q., X.W., A.H.H., E.U., and Y.X. Writing—

original draft: Y.J., Y.Z., and P.L. Writing—review & editing: I.D., W.Y.K., K.I.P., and H. S.E. Funding acquisition: N.V.D., G.W., C.M.P., W.Y.K., H.S.E., and P.L. Resources: X. W., N.V.D., G.W., and C.M.P. Supervision: W.Y.K., H.S.E., and P.L.

## Additional information

**Competing interests:** H.S.E. is a founder of Meryx (a UNC startup) that is developing small molecule inhibitors for MERTK. H.S.E. and X.W. own stock in Meryx. The other authors declare no competing interests.

