## [Peer Review File · Nature Communications]

Reviewers' comments:

Reviewer #1 (Remarks to the Author):

This manuscript "SAV1 is an endogenous Akt inhibitor: MERTK-mediated Novel Site Akt phosphorylation alleviates suppression" by Jiang et al. presented a very nice story with several interesting discoveries including (1) the SAV1-Akt biochemical/functional interactions, (2) the identification of specific binding of SAV1 to the conserved Akt PRY domain, (3) the connection between SAV1 level and Akt activation in RCC, (4) the identification of MERTK as the kinase phosphorylates Akt, and (5) the diminished oncogenic potential of Akt Y26F mutant. A few comments to help the authors strengthen their conclusions.

1) ACHN is a papillary RCC (NCom, 2016, 28489074). Hence, the SAV1-AKT dysfunction might also occur in pRCC. The low SAV1 expression in KIRP supports this notion, too.

2) Figure S2A legend and labels are extremely unclear. What is the y axis?

3) The analysis between SAV1 and AKT signaling should be examined in KIRC and KIRP TCGA datasets. (Fig S2).

4) Page 6, "supplementary Fig. S3j" should be S2j

5) Fig. 2a cabozantinib which inhibits AXL (TAM family protein as MERTK) also enhances SAV/Akt interaction but to a lesser degree. Please examine the role of AXL in this context.

6) Page 7. Please explain the inclusion of GAS6 and PROS in the study.

7) Fig. S3f is not convincing, please quantify.

8) Fig. S7c is confusing. At 30min treatment time point with PV, the Akt-pY26 reduced to a level similar to 5 min treatment and yet the interactions with SAV1 were dramatically different.

Reviewer #2 (Remarks to the Author):

Major claims:

In this paper, Jiang et al identify SAV1 as an endogenous inhibitor of Akt. They examine in detail the binding mechanism via PH-WW interaction. Furthermore, cancer associated mutations were identified that impair SAV1 binding. In addition, phosphorylation of the Akt1-PH domain by MERTK releases SAV1 binding. The degree of SAV1 binding affects tumorigenesis in mouse models, sensitivity to MERTK inhibitors, and (possibly) prognosis in patients.

Novelty:

The above mentioned findings are novel and will be of broad interest to the cancer field, given the crucial role of Akt in cancer cell signaling.

Influence on thinking in the field:

This paper has the potential to change thinking in the cancer biology and clinical oncology field, because:

- it shows that, besides the canonical pathway of Akt activation, there is an additional pathway involving dislodgement of the SAV1 inhibitor
- it indicates that sensitivity to therapeutic kinase inhibitors will not only be influenced by the status of the target kinase (in case MERTK), but also by the possible association of its substrate (AKT) with an endogenous inhibitor (SAV1).

Specific comments:

- page 3: the authors mention that "whether and how Akt binding proteins modulate Akt activity in cancer ... remains to be determined." However, there is a plethora of information on Akt binding proteins and their effect on Akt activity. The authors could e.g. refer to this list:

<https://www.cellsignal.com/contents/resources-reference-tables/pi3k-akt-binding-partners-table/science-tables-akt-binding>

- Figure S2: I am not very convinced by the inverse correlation data between SAV1 expression and Akt activity in tumour tissue or clinical prognosis. I think that MERTK activity needs to be brought

into this equation, since the Akt inhibitory effect of high SAV1 expression may still be counteracted by high MERTK activity. I think this is the reason why the results of the blots (S2b) and of the correlations in the tumor stages (S2c) are a bit variable and not very conclusive.

- the authors stimulate cells with pervanadate (PV) as a general stimulator of tyrosine kinase signaling. Why don't they stimulate their cells with GAS6, which is the ligand of MERTK? This could make the results of Fig2k more connected to MERTK.

- colony formation results in Fig 3o and 3p would be strengthened if they would be complemented by mouse experiments with tumor cells expressing the same set of mutants.

- it would be important to find out the relative contribution of EGF vs MERTK to Akt activation in SAV+ or SAV- cell lines. A nice experiment could be to collect a set of cell lines and test them for expression, protein level and activity of EGFR, MERTK, Akt and SAV1, in order to test whether the capacity of the EGFR to activate Akt is correlated to the degree of SAV1 expression and MERTK activity.

Minor comment:

- the readability of the manuscript is sometimes reduced due to lack of correct punctuation (sometimes comma's have been forgotten).

Conclusion:

Given the novelty and importance of this work, I think this manuscript can be considered for publication if the authors would submit a revised version taking into account the above mentioned suggestions for additional experiments.

Reviewer #3 (Remarks to the Author):

This paper by Jiang and colleagues describes a novel mechanism of Akt negative regulation based on the interaction between Akt1 and the WW-domain containing protein SAV1. Binding of SAV1 to a proline-tyrosine motif within the PH domain of Akt1 inhibits Akt upstream activation. Akt can be released from SAV1 binding by MERTK-mediated phosphorylation at Y26 within the PH domain, which restores responsiveness to canonical PI3K signaling. The authors also identify cancer-associated SAV1 mutations with impaired ability to bind Akt, which may enhance Akt oncogenic functions. As such, by identifying a new layer of Akt regulation in patho-physiological conditions like RCC, this study is potentially interesting as it may pave the way for the design of new therapeutic strategies for counteracting Akt-driven oncogenesis.

Although the main take-home message of the manuscript is reasonably well supported by experimental data, there are several points that require additional validation, and some inconsistencies that require clarification prior to publication.

- The model proposed by the authors implies that SAV1 silencing/deletion should enhance Akt activation in response to upstream agonists. This model is inferred mostly by analyzing Akt phosphorylation in cells under steady-state growth conditions and not after acute stimulation with Akt agonists. Akt phosphorylation at the steady state results from the balance between positive (PDK1, mTORC2) and negative regulators (e.g. PHLPP and PP2A phosphatases). A direct analysis of the effects of SAV1 manipulation (silencing/deletion/overexpression) on Akt activation should be provided. This should be done by comparing the kinetics of Akt phosphorylation (both at Thr308 and Ser473) in response to acute EGF/insulin stimulation after starvation at various time points in different cell lines.
- The same experimental strategy should also be applied to the analysis of Akt phosphorylation at Thr308 and Ser473 following MERTK pharmacological and/or genetic manipulation in relevant cell lines.
- In Fig. 1g, DLD1-Akt1/2-/- cells show significant levels of endogenous Akt proteins in EV transfected cells based on Akt total immunoblots (is Akt3 the residual Akt protein?). However, Akt phosphorylation is detected only in the presence of ectopically expressed HA-Akt1. Since the levels

of total Akt proteins do not differ significantly between cells transfected with EV and Akt mutants, the authors should provide an explanation for this observation.

- In several experiments (like for instance those shown in Fig 1g and 1j), manipulation of SAV1 expression has a much more pronounced effect on Akt phosphorylation at Thr308 as compared to Ser 473. Why?
- In several experiments (e.g. Fig 1j, 1l, S3 a-i), the changes in phosphorylation levels of Akt at Thr308 induced by SAV1 manipulation are not always reflected by parallel changes in the phosphorylation of Akt downstream signaling molecules (S6K-pT389, GSK-pS9). Clarification of this point is crucial for establishing the effects on Akt signaling outputs.
- In some experiments, Akt protein levels are monitored by pan-Akt immunoblotting (e.g. 1g, 1j, 1l, etc.), whereas in other experiments, Akt expression is monitored by Akt1 immunoblotting (e.g. 1h, 2b, 2c, 2e, 2f, etc.). Moreover, Akt activation in some experiments is monitored via Thr308 phosphorylation analysis (e.g. in Fig 2k), whereas in other experiments this is done by analysis of phosphorylation at Ser 473 (e.g. Fig 2l), or both (e.g., Fig 2b, 2c). Authors should use uniform indicators of Akt expression/activity across experiments, or they should explain why they are using one specific marker in particular experimental settings.
- The use of PV as a surrogate inducer of MERTK activity may be misleading, especially in respect to the effects on Akt Thr308 and Ser 473 phosphorylation. In fact, in Fig 2i, the PV-induced phosphorylation of Akt at Y26 is completely abrogated by a concomitant MERTK inhibition whereas phosphorylation at Thr308 is only partially inhibited. This indicates that PV has MERTK-independent effects on Akt phosphorylation. Similarly, PV induces Akt phosphorylation at Ser473 even in the presence of MERTK inhibitor that totally abrogates Akt phosphorylation at Y26 (Fig. 2l). Authors should comment on this.
- Which are the effects of the expression of a constitutively active MERTK mutant (Wu Y et al., Journal of Cell Science, 2005) on Akt phosphorylation (Y26, Thr308, Ser473) in response to Akt upstream agonists in RCC cell lines?
- Which are the roles of the SAV1-MERTK axis in respect to Akt signaling activation in cancer cells expressing constitutively active PI3K or EGFR mutants?

Minor points:

- 1) The quality of Akt-pThr308 immunoblotting of Fig 1h is very poor. The relevant bands have too much contrast and are half-visible. Please replace with better quality images.
- 2) In Fig S4, the labeling of the panel c is missing (there are two "b" labels).
- 3) The legend of panel f of Fig S8 is missing.

Point-to-Point Responses to the Reviewers' Critiques (NCOMMS-18-21029)

We deeply appreciate the thorough analysis and constructive suggestions provided by the three reviewers during the review of our manuscript at *Nature Communications*. The thoughtful comments have been very helpful in guiding us to further improve our study. With this round of extensive revision, as described in more detail below, we hope that the editor and the reviewers concur with us that we have fully addressed all the raised concerns in a satisfactory manner and, substantially strengthened our paper. Therefore, we believe that the revised manuscript is now suitable for publication in *Nature Communications*.

Reviewer: 1

This manuscript "SAV1 is an endogenous Akt inhibitor: MERTK-mediated Novel Site Akt phosphorylation alleviates suppression" by Jiang et al. presented a very nice story with several interesting discoveries including (1) the SAV1-Akt biochemical/functional interactions, (2) the identification of specific binding of SAV1 to the conserved Akt PRY domain, (3) the connection between SAV1 level and Akt activation in RCC, (4) the identification of MERTK as the kinase phosphorylates Akt, and (5) the diminished oncogenic potential of Akt Y26F mutant.

Response: We thank the reviewer for recognizing the potential impacts of our study. We also appreciate the constructive comments from the reviewer to help us further improve our manuscript.

A few comments to help the authors strengthen their conclusions.

1) ACHN is a papillary RCC (NCom, 2016, 28489074). Hence, the SAV1-AKT dysfunction might also occur in pRCC. The low SAV1 expression in KIRP supports this notion, too.

Response: We thank the reviewer for raising this excellent point. We fully agree with the reviewer that ACHN is a papillary RCC line. Following the reviewer's suggestion, we have examined the correlation between SAV1 expression (mRNA levels) and Akt-pT308 (RPPA data) in TCGA-KIRP datasets and no significant correlation was observed (revised Supplementary Fig. S2k). This may be partially due to that unlike ccRCC patient samples with clear disease stages (revised Supplementary Figs. S2c to S2f), pRCC patient samples are lacking the disease stage characterization with only 210 cases included for all stages (revised Supplementary Fig. S2g). This analysis would be more statistically robust if more samples were included. In addition, a recent report (Wang et al. Comprehensive Molecular Characterization of the Hippo Signaling Pathway in Cancer. *Cell Reports*. 2018. 25: 1304-1317) indicates that *SAV1* deletion is more enriched in KIRC compared with KIRP patients. Given that our studies focus more on ccRCC, the exact roles of the SAV1-Akt dysfunction in KIRP would warrant in-depth investigations with the availability of more sequenced patient samples. Moreover, considering that both SAV1 deficiency and MERTK activation contribute to Akt activation, it is also plausible that combined SAV1 deficiency and MERTK activation may correlate with Akt activation in pRCC, which warrants further investigations as well if more MERTK activation markers (such as phospho-Y867-MERTK signals) can be included in pRCC RPPA analyses.

2) Figure S2A legend and labels are extremely unclear. What is the y axis?

Response: We apologize this labeling deficiency. In the revised manuscript we have clearly labeled both X-axis and Y-axis. Y-axis labels the mRNA expression levels of SAV1.

3) The analysis between SAV1 and AKT signaling should be examined in KIRC and KIRP TCGA datasets. (Fig S2).

Response: We thank the reviewer for raising this outstanding point. Following the reviewer's suggestion, we have included analyses of SAV1 expression and Akt-pT308 in KIRP dataset (revised Supplementary Fig. S2g), where we didn't observe a significant correlation. In addition to Akt-pT308, we also analyzed the correlation of protein levels of other Akt signaling members with SAV1 mRNA expression in both stage IV KIRC and KIRP TCGA datasets. Interestingly, there is a moderate inverse correlation between SAV1 expression and GSK3-S9 phosphorylation (as an Akt substrate in part reflecting Akt-pT308-mediated Akt activity) in KIRC (revised Supplementary Fig. S2g) but not KIRP (revised Supplementary Fig. S2m) datasets. Although a moderate inverse correlation between SAV1 expression and Akt-pT308 in stage IV KIRC dataset (revised Supplementary Fig. S2c), no significant correlation was observed between SAV1 expression and Akt-pS473 in neither stage IV KIRC (revised Supplementary Fig. S2h) nor KIRP (revised Supplementary Fig. S2l) datasets. Moreover, no significant correlation was observed between SAV1 expression and other upstream Akt activating enzymes including mTOR (revised Supplementary Fig. S2i) and PDK1 (revised Supplementary Fig. S2j) in KIRC dataset. No significant correlation between SAV1 expression and mTOR (revised Supplementary Fig. S2n) or PDK1 (revised Supplementary Fig. S2o) was observed in KIRP dataset. Cumulatively, these analyses suggest that SAV1 may exert a role in negatively regulating Akt-pT308 without significantly affecting canonical Akt upstream activators such as mTOR and PDK1. Although our cell line analysis does reveal a SAV1 dependent interference in mTOR and PDK1, this correlation is not strong enough to be observed in clinical data with its additional complexities.

4) Page 6, "supplementary Fig. S3j" should be S2j

Response: On page 6 we performed experiments as included in "supplementary Fig. S3j" to examine whether SAV1 suppresses Akt in a Hippo-signaling independent manner by using either *MST1/2*^{-/-} or *LATS1/2*^{-/-} cells. The referred figure here is "Supplementary Fig. S3j".

5) Fig. 2a cabozantinib which inhibits AXL (TAM family protein as MERTK) also enhances SAV/Akt interaction but to a lesser degree. Please examine the role of AXL in this context.

Response: We thank the reviewer for raising this great point and agree that the broader based MET and TAM family inhibitor showed an intermediate effect (Fig. 2a). And we also agree with the reviewer that it is important to dissect the roles of other TAM family members in regulating Akt. To examine the role of Axl in Akt activation, using multiple independent sgRNAs, we found that deleting endogenous *Axl* did not significantly reduce Akt-pT308 signals in both RCC4 cells and UMRC2 cells (revised Supplementary Figs. S4h and S4i), as compared to the effects of control deletion. Loss of *Axl* in RCC4 cells didn't significantly retard growth of RCC4 cells *in vitro* (revised Supplementary Figs. S4l and S4m). These data suggest that Axl may not play a critical role in modulating Akt activation in these RCC cells. Given that the TAM family of kinases include Tyro3, Axl and MERTK, to further investigate roles of other TAM kinases in addition to MERTK and Axl, we also deleted endogenous *Tyro3* by CRISPR in both RCC4 and Caki-I cells. Notably, no significant changes on Akt phosphorylation were observed (revised Supplementary Figs. S4j and S4k), suggesting that Tyro3 may not be a major kinase phosphorylating Akt in these RCC cell lines examined. Data from knock out mouse kidney cells were somewhat less stark but the cell line elimination of *Axl* and *Tyro3* substantiates the Akt1-Y26 phosphorylation as a major MERTK effect. Together, these data support that MERTK, but not Axl nor Tyro3, plays a more specific role in phosphorylating and activating Akt in RCC.

Given that cabozantinib inhibits a group of receptor tyrosine kinases (RTKs) including MET, VEGFR, RET, Axl, Kit and FLT3, the moderate increase of SAV1 binding to Akt1 upon cabozantinib administration may be derived from other RTKs than Axl. However, compared with MERTK inhibitors (UNC2025 and UNC4241), the increase observed is quite minor (revised Fig. 2a). Although it is plausible that other RTKs are also involved

in this regulation, our data suggest that the major kinase responsible for Akt phosphorylation that attenuates SAV1 binding is MERTK. Following the reviewer's suggestion, we have toned down our statement that other RTKs may also play a role in regulating SAV1 binding to Akt, although MERTK is the major candidate in our current search and the focus for this study.

6) Page 7. Please explain the inclusion of GAS6 and PROS in the study.

Response: We thank the reviewer for raising this point. While detailed binding studies of the lipid (PtdSer) protein complex are lacking, much data points to the optimal activation of MERTK is achieved by an extracellular lipid-protein complex composed of both PtdSer (phosphatidylserine) and Gas6 or Protein S (PROS) (Graham et al. The TAM family: phosphatidylserine-sensing receptor tyrosine kinases gone awry in cancer. *Nature Reviews Cancer*. 2014. 14: 769–785). While it has been observed that PtdSer itself is strong enough to trigger MERTK activation in some settings due to the presence of GAS6 or PROS (Rankin et al. Direct regulation of Gas6/AXL signaling by HIF promotes renal metastasis through SRC and MET. *PNAS*. 2014. 111(37): 13373-13378). In our experimental system, we found that PtdSer alone was able to promote MERTK activation (Fig. 2h), presumably due to the presence of endogenous Gas6 that RCC4 cells produce measured by RT-PCR (revised Supplementary Figs. S6b and S6c). To avoid confusions for future readers, we have included a reference in this section to provide more background and removed the PROS from the revised data that is not very relevant to our current study.

7) Fig. S3f is not convincing, please quantify.

Response: We thank the reviewer for raising this point. We think the reviewer is referring to the revised Fig. S5f (original Fig. S4f) instead of original Fig. S3f, given that the signal changes in Fig. S3f is dramatic. Following the reviewer's suggestion, we have quantified the Akt-pT308 signals in the revised Fig. S5f and included it as the revised Supplementary Fig. S5g. The quantification results support a role of MERTK in promoting Akt-pT308 phosphorylation in these TAM knockout mouse models, given that loss of *mertk* leads to more than a 5-fold decrease in Akt-pT308 signals.

8) Fig. S7c is confusing. At 30min treatment time point with PV, the Akt-pY26 reduced to a level similar to 5 min treatment and yet the interactions with SAV1 were dramatically different.

Response: We thank the reviewer for raising this point and we agree with the reviewer that at the original 30min treatment time point, although the Akt1-pY26 signals were reduced to a level comparable to 5min treatment, the increase of Akt1 binding to SAV1 was more dramatic than that at 5min. There may be multiple reasons behind this observation. For example, it might be in part due to that more cytoplasmic Akt is available 30min post-PV treatment for SAV1 binding, or SAV1 is differentially phosphorylated at 5min or 30min post-PV treatment that may affect its binding to Akt. To alleviate these concerns, we have repeated this experiment and observed that in this repeated experiment at early time points, Akt1-Y26 phosphorylation inversely correlated with SAV1 binding to Akt (revised Supplementary Fig. S9c), which supports our original conclusions. In addition, to examine a direct role of MERTK activation in regulating this process, we co-expressed a constitutive active MERTK (Fc-MERTK) and found that Fc-MERTK expression led to reduced SAV1 binding to Akt1 (revised Supplementary Fig. S9d).

Reviewer #2 (Remarks to the Author):

Major claims:

In this paper, Jiang et al identify SAV1 as an endogenous inhibitor of Akt. They examine in detail the binding mechanism via PH-WW interaction. Furthermore, cancer associated mutations were identified that impair SAV1 binding. In addition, phosphorylation of the Akt1-PH domain by MERTK releases SAV1 binding. The degree of SAV1 binding affects tumorigenesis in mouse models, sensitivity to MERTK inhibitors, and (possibly) prognosis in patients.

Novelty:

The above mentioned findings are novel and will be of broad interest to the cancer field, given the crucial role of Akt in cancer cell signaling.

Influence on thinking in the field:

This paper has the potential to change thinking in the cancer biology and clinical oncology field, because:

- it shows that, besides the canonical pathway of Akt activation, there is an additional pathway involving dislodgement of the SAV1 inhibitor
- it indicates that sensitivity to therapeutic kinase inhibitors will not only be influenced by the status of the target kinase (in case MERTK), but also by the possible association of its substrate (AKT) with an endogenous inhibitor (SAV1).

Response: We thank the reviewer for recognizing the novelty and importance of our studies and providing constructive suggestions to further improve our manuscript.

Specific comments:

- page 3: the authors mention that "whether and how Akt binding proteins modulate Akt activity in cancer ... remains to be determined." However, there is a plethora of information on Akt binding proteins and their effect on Akt activity. The authors could e.g. refer to this list: <https://www.cellsignal.com/contents/resources-reference-tables/pi3k-akt-binding-partners-table/science-tables- akt-binding>

Response: We thank the reviewer for pointing out this critical information and fully agree with the reviewer that 34 Akt binding partners are included in the table from CST as the reviewer kindly pointed out. This list includes Akt modifying enzymes (such as PHLPP, SIRT2, PP1 and others), Akt substrates (such as Skp2, p27, PEA-15 and others) and other binding proteins that are not modified by Akt or modifying Akt (such as GAPDH, Jade-1 and others). In our study, we focus on the third class of Akt binding partners-non-enzyme targets that have not been well-characterized and targeted for cancer therapy. Thus, the actually number of this type of Akt binding proteins is small (less than 10). Following the reviewer's suggestion and to avoid possible confusions to future readers, we have revised the text to clarify our statement on page 3 of the revised manuscript.

- **Figure S2: I am not very convinced by the inverse correlation data between SAV1 expression and Akt activity in tumour tissue or clinical prognosis. I think that MERTK activity needs to be brought into this equation, since the Akt inhibitory effect of high SAV1 expression may still be counteracted by high MERTK activity. I think this is the reason why the results of the blots (S2b) and of the correlations in the tumor stages (S2c) are a bit variable and not very conclusive.**

Response: We fully agree with the reviewer that the MERTK activity would also need to be taken into consideration. Following the reviewer's suggestion, in RCC patient samples, we also monitored the MERTK activity by antibodies against characterized MERTK downstream targets (given that the phospho-Y867-MERTK antibody cannot detect endogenous MERTK phosphorylation in cell extracts), including p38-

pT180/pY182 (Guttridge et al. Mer receptor tyrosine kinase signaling: prevention of apoptosis and alteration of cytoskeletal architecture without stimulation or proliferation. *The Journal of Biochemical Chemistry*. 2002. 277(27):24057-24066) and ERK-pS42/pT44 (Schlegel et al. MERTK receptor tyrosine kinase is a therapeutic target in melanoma. *Journal of Clinical Investigation*. 2013. 123(5): 2257–2267) (revised Supplementary Fig. S2b). Indeed, in some RCC patient samples with no significant SAV1 expression changes, MERTK activation correlates with increased Akt-pT308 signals. These data suggest that either SAV1 loss, or MERTK hyperactivation, or both of the mechanisms, would lead to Akt hyperactivation.

In addition, we have also examined correlations between Akt-pT308 and MERTK activity (by MEK-pS217/pS221 phosphorylation (Guttridge et al. Mer receptor tyrosine kinase signaling: prevention of apoptosis and alteration of cytoskeletal architecture without stimulation or proliferation. *The Journal of Biochemical Chemistry*. 2002. 277(27):24057-24066)) using TCGA KIRC RPPA dataset and found that only in stage IV patients there is a moderate positive correlation between these two events (revised Supplementary Figs. S5n to S5q). Given to the small sample size (81 patients), no statistical significance could be obtained, and more patients samples are needed to determine whether MERTK activity correlates with Akt activation in KIRC patients. Nonetheless, we fully agree with the reviewer that both SAV1 loss and MERTK activation contribute to Akt-T308 phosphorylation and Akt activation.

- the authors stimulate cells with pervanadate (PV) as a general stimulator of tyrosine kinase signaling. Why don't they stimulate their cells with GAS6, which is the ligand of MERTK? This could make the results of Fig2k more connected to MERTK.

Response: We thank the reviewer for raising this point. The optimal activation of MERTK is achieved by an extracellular lipid-protein complex composed of both PtdSer (phosphatidylserine) and Gas6 or Protein S (PROS) (Graham et al. The TAM family: phosphatidylserine-sensing receptor tyrosine kinases gone awry in cancer. *Nature Reviews Cancer*. 2014. 14: 769–785). In our original manuscript, in addition to PV, we have also treated RCC4 cells with PtdSer to stimulate MERTK activation (Fig. 2h), in the presence of endogenous GAS6 (revised Supplementary figs. S6b and S6c). It is notorious that most of GAS6 obtained commercially does not reliably trigger MERTK activation, presumably due to lack of certain post-translational modifications on GAS6 that are critical for GAS6 to serve as a high-affinity MERTK ligand. Following the reviewer's suggestion, we have treated RCC4 cells with different batches of GAS6 obtained from different resources and found that GAS6 synthesized by the Earp lab displayed a greater effect in triggering MERTK activation (as evidenced by Axl-pY702 signals, revised Supplementary Figs. S6d). Importantly, administration of GAS6 alone to RCC4 cells (presumably PtdSer is available from apoptotic cells in cell culture) triggered Akt1-Y26 and Akt-T308 phosphorylation (revised Supplementary Fig. S6d). These data support a role of GAS6 in promoting Akt activation mediated by MERTK activation.

- colony formation results in Fig 3o and 3p would be strengthened if they would be complemented by mouse experiments with tumor cells expressing the same set of mutants.

Response: We thank the reviewer for bringing up this suggestion and fully agree with reviewer that a mouse experiment will be more convincing. Following the reviewer's suggestion, we have performed a mouse xenograft experiment using 786-O cells reconstituted with either EV, WT or various SAV1 cancer mutations (revised Supplementary Fig. S12d). Consistent with results observed from previous colony formation assays (revised Supplementary Figs. S12f and S12g), expression of WT-SAV1 greatly suppressed Akt-pT308 and subsequently retarded 786-O tumor growth xenografted subcutaneously in nude mice, while cancerous SAV1-H220Q and SAV1-R233Q expressing 786-O cells formed bigger tumors than the control cells infected with empty vector (revised Figs. 3o to 3q and revised Supplemental Fig. S12e). These data support the notion that

SAV1 suppresses Akt activation, therefore leading to reduced tumor growth; while cancer patient-derived SAV1-WW1 mutations (including SAV1-H220Q and R223Q) gain oncogenicity through bypass binding and suppressing Akt activity, thus resulting in elevated tumor growth.

- it would be important to find out the relative contribution of EGF vs MERTK to Akt activation in SAV⁺ or SAV⁻ cell lines. A nice experiment could be to collect a set of cell lines and test them for expression, protein level and activity of EGFR, MERTK, Akt and SAV1, in order to test whether the capacity of the EGFR to activate Akt is correlated to the degree of SAV1 expression and MERTK activity.

Response: We fully agree with the reviewer that it is important to determine the relative contributions of EGFR and MERTK in Akt activation in SAV1^{+/+} and SAV1^{-/-} RCC cell lines. Following the reviewer's suggestion, in a panel of RCC cell lines in the revised Fig. 1i that include both SAV1^{+/+} and SAV1^{-/-} cells, we have obtained further immuno-blots for EGFR, pY1068-EGFR (as evidence for EGFR activity), pT180/pY128-p38 (an indicator for MERTK activity, Guttridge et al. Mer receptor tyrosine kinase signaling: prevention of apoptosis and alteration of cytoskeletal architecture without stimulation or proliferation. *The Journal of Biochemical Chemistry*. 2002. 277(27): 24057-24066), Akt-pT308, Akt-pS473 and Akt1. As demonstrated in the revised Fig. 1i, we didn't observe a significant contribution of EGFR activation in Akt phosphorylation in neither a SAV1 expression nor MERTK expression/activity dependent manner. Considering that this is a correlation study and these cell lines contain many other genetic changes than the ones we have examined, we are of the opinion that it is difficult to examine whether EGFR and MERTK contributes to Akt activation differentially depending on SAV1 expression.

In addition, as Reviewer#3 pointed out, to further evaluate effects of constitutive EGFR hyper-activation mutations in regulating the MERTK/Akt/SAV1 signaling, we ectopically expressed cancerous hot-spot hyperactivation mutation EGFR-L858R, with its WT counterpart as a control in SAV1-depleted RCC4 or Caki-I cells. We observed that although depletion of endogenous SAV1 led to increased Akt-pT308 signals upon either EGF or insulin stimulation, expression of EGFR-L858R in SAV1-depleted RCC cells didn't result in elevated Akt-pT308 upon EGF stimulation (revised Supplementary Fig. S10i). To further examine whether EGFR hyperactivation can compensate for the loss of *MERTK*, we ectopically expressed EGFR-L858R in MERTK-deleted RCC4 cells and found that hyperactivation of EGFR (revised Supplementary Fig. S10j) could not rescue MERTK-deletion induced decreases in Akt-pT308. These data suggest that MERTK may be one of the kinases through which SAV1-mediated Akt suppression is released.

Minor comment:

- the readability of the manuscript is sometimes reduced due to lack of correct punctuation (sometimes comma's have been forgotten).

Response: We apologize for the lack of punctuation in our originally submitted manuscript. We have sought help from our colleagues to improve the readability of the manuscript during the revision process. Hope the reviewer concurs with us that the revised manuscript reads much better. We sincerely thank the reviewer for this suggestion.

Conclusion:

Given the novelty and importance of this work, I think this manuscript can be considered for publication if the authors would submit a revised version taking into account the above mentioned suggestions for additional experiments.

Response: We sincerely thank the reviewer for many constructive suggestions that have been very helpful in guiding us during the revision process to further improve our manuscript. We hope the reviewer concurs with us that our revised manuscript with inclusion of new experimental evidence has satisfactorily addressed the concerns raised by the reviewer.

Reviewer #3 (Remarks to the Author):

This paper by Jiang and colleagues describes a novel mechanism of Akt negative regulation based on the interaction between Akt1 and the WW-domain containing protein SAV1. Binding of SAV1 to a proline-tyrosine motif within the PH domain of Akt1 inhibits Akt upstream activation. Akt can be released from SAV1 binding by MERTK-mediated phosphorylation at Y26 within the PH domain, which restores responsiveness to canonical PI3K signaling. The authors also identify cancer-associated SAV1 mutations with impaired ability to bind Akt, which may enhance Akt oncogenic functions. As such, by identifying a new layer of Akt regulation in patho- physiological conditions like RCC, this study is potentially interesting as it may pave the way for the design of new therapeutic strategies for counteracting Akt-driven oncogenesis.

Response: We thank the reviewer for recognizing the importance of our studies and providing constructive suggestions to further improve our manuscript.

Although the main take-home message of the manuscript is reasonably well supported by experimental data, there are several points that require additional validation, and some inconsistencies that require clarification prior to publication.

- **The model proposed by the authors implies that SAV1 silencing/deletion should enhance Akt activation in response to upstream agonists. This model is inferred mostly by analyzing Akt phosphorylation in cells under steady-state growth conditions and not after acute stimulation with Akt agonists. Akt phosphorylation at the steady state results from the balance between positive (PDK1, mTORC2) and negative regulators (e.g. PHLPP and PP2A phosphatases). A direct analysis of the effects of SAV1 manipulation (silencing/deletion/overexpression) on Akt activation should be provided. This should be done by comparing the kinetics of Akt phosphorylation (both at Thr308 and Ser473) in response to acute EGF/insulin stimulation after starvation at various time points in different cell lines.**

Response: We thank the reviewer for raising this great suggestion and fully agree with the reviewer that measurements of Akt activation kinetics at both pT308 and pS473 upon physiological stimulations will provide more detailed understanding of roles SAV1 in governing Akt activation. Following the reviewer's suggestions, to analyze the effects of SAV1 manipulation on Akt activation, we first ectopically expressed SAV1-WT in SAV1^{-/-}786-O cells (no SAV1 expression can be detected in this cell line, revised Fig. 1i) and found that upon EGF stimulation, both Akt-pT308 and Akt-pS473 were dramatically reduced at all time points examined (Supplementary Fig. S4a), suggesting that SAV1 suppresses Akt activation upon EGF stimulation. Furthermore, upon EGF stimulation, depletion of endogenous SAV1 led to elevated Akt-pT308 signals in both RCC4 cells (revised Supplementary Fig. S4b) and Caki-I cells (revised Supplementary Fig. S4c), compared with corresponding shscramble control cells. While the changes in Akt-pS473 signals are minor. Similarly, SAV1 depletion in RCC4 cells also resulted in significantly elevated Akt-T308 phosphorylation upon insulin stimulation (Supplementary Fig. S4d). These data cumulatively suggest that SAV1 suppresses Akt activation, especially Akt-T308 phosphorylation, under both steady state conditions and acute EGF/insulin stimulation conditions. In new supplemental data (see below), we also show that constitutive activation of MERTK, which would relieve SAV1 PH domain Akt binding, enhances Akt activation by both EGF and insulin.

- **The same experimental strategy should also be applied to the analysis of Akt phosphorylation at Thr308 and Ser473 following MERTK pharmacological and/or genetic manipulation in relevant cell lines.**

Response: We fully agree with the reviewer that the examining the kinetics of Akt phosphorylation in response to growth factors upon MERTK manipulation will further facilitate our understanding of the roles of MERTK

in Akt activation. Following the reviewer's suggestions, to specifically determine the role of MERTK, but not other TAM kinases that UNC2025 and UNC4241 may also inhibit, we have examined how MERTK deficiency affects Akt-T308 phosphorylation responding to EGF or insulin stimulation. Notably, as shown in the revised Supplementary Fig. S6g and S6h, loss of *MERTK* significantly reduced Akt-T308 phosphorylation in responding to both EGF and insulin stimulation, respectively. Moreover, ectopic expression of a constitutive active MERTK (Fc-MERTK, Kasikara et al. Phosphatidylserine Sensing by TAM Receptors Regulates AKT-Dependent Chemoresistance and PD-L1 Expression. *Molecular Cancer Research*. 2017. DOI: 10.1158) in UMRC6 cells led to increased and sustained Akt-pT308 phosphorylation upon either EGF (revised Supplementary Fig. S6g) or insulin (revised Supplementary Fig. S6h) stimulation. These data suggest that MERTK facilitates Akt activation under both steady state conditions and acute EGF/insulin stimulation conditions.

• In Fig. 1g, DLD1-Akt1/2^{-/-} cells show significant levels of endogenous Akt proteins in EV transfected cells based on Akt total immunoblots (is Akt3 the residual Akt protein?). However, Akt phosphorylation is detected only in the presence of ectopically expressed HA-Akt1. Since the levels of total Akt proteins do not differ significantly between cells transfected with EV and Akt mutants, the authors should provide an explanation for this observation.

Response: We thank the reviewer for raising this concern and sincerely apologize for not indicating the total Akt bands well. We agree with the reviewer that the residue signal may be from Akt3. To avoid confusions and following the reviewer's suggestion below, we have replaced all total Akt blots with Akt1 blots, including Fig. 1g. In the newly obtained Akt1 blot, it is clearly demonstrated that there is no signal in the EV sample, consistent with its *Akt1/2^{-/-}* background.

• In several experiments (like for instance those shown in Fig 1g and 1j), manipulation of SAV1 expression has a much more pronounced effect on Akt phosphorylation at Thr308 as compared to Ser 473. Why?

Response: We thank the reviewer for this important concern and agree with the reviewer that there is crosstalk between pS473-Akt and pT308-Akt that should be taken into consideration when interpreting our experimental results. Indeed, when SAV1 is depleted, we observed more dramatic increase on Akt-pT308 than Akt-pS473 (revised Figs. 1h, 1j and 1i). Importantly, it has been previously documented that PDK1-mediated pT308-Akt plays a relatively more important role in maintaining Akt kinase activity than mTORC2-mediated pS473-Akt (Aoki et al. The Akt kinase: Molecular determinants of oncogenicity. *PNAS*. 1998. 95: 14950-14955; Rodrik-Outmezguine VS et al. mTOR kinase inhibition causes feedback-dependent biphasic regulation of AKT signaling. *Cancer Discovery*. 2011. 1: 248-59). In addition, as mTORC2, the major Akt-pS473 kinase resides at both ribosomes (Zinzalla et al. Activation of mTORC2 by association with the ribosome. *Cell*. 2011. 144(5):757-68) and plasma membrane (Liu et al. PtdIns(3,4,5)P3-dependent Activation of the mTORC2 Kinase Complex. *Cancer Discovery*. 2015. doi: 10.1158/2159-8290.), it is plausible that SAV1 binding to Akt may largely attenuate Akt binding to plasma membrane-located mTORC2, but not ribosomal mTORC2, thus SAV1 may not fully suppress Akt-S473 phosphorylation. Given that to date we cannot distinguish differentially located mTORC2s, and Akt-T308 phosphorylation accounts for more than 70% of Akt activity, we focus on understanding how SAV1 or MERTK governs the Akt-T308 phosphorylation in this study. However, we agree with the reviewer that changes in pS473-Akt will likely to be observed given its crosstalk with pT308-Akt but may be relatively moderate in our experimental conditions.

• In several experiments (e.g. Fig 1j, 1l, S3 a-i), the changes in phosphorylation levels of Akt at Thr308 induced by SAV1 manipulation are not always reflected by parallel changes in the phosphorylation of

Akt downstream signaling molecules (S6K-pT389, GSK-pS9). Clarification of this point is crucial for establishing the effects on Akt signaling outputs.

Response: We thank the reviewer for bringing up this question. We agree with the reviewer that S6K-pT389 is a downstream signaling molecule of Akt; however, Akt is only one of the upstream signaling for mTORC1 activation under growth stimulation conditions. In addition, mTORC1 is also regulated by AMPK upon energy stress (Gwinn, et al. AMPK phosphorylation of raptor mediates a metabolic checkpoint. *Molecular Cell*. 2008. 30(2):214-26), by ERK under Ras activation conditions (Carriere et al. ERK1/2 Phosphorylate Raptor to Promote Ras-dependent Activation of mTOR Complex 1 (mTORC1). *Journal of Biochemical Chemistry*. 2011. 286, 567-577), and Rag GTPases upon amino acid stimulation conditions (Efeyan et al. Rag GTPase-mediated regulation of mTORC1 by nutrients is necessary for neonatal autophagy and survival. *Nature*. 2013. 493(7434): 679–683). Thus, deficiency in Akt signaling may not always lead to significant changes in mTORC1 activity, which is cellular condition dependent. To this end, we observed that in DLD1 cells, genetic deletion of both *Akt1* and *Akt2* led to almost abolished Akt phosphorylation on both T308 and S473, while it only had minimal effects on S6K-pT389 (Figure R1). These data suggest that mTORC1 activity can be partially uncoupled from Akt signaling.

Figure R1. Western blot analyses of mTOR signaling in DLD1-*Akt1/2*^{-/-} cells indicate that loss of *Akt1/Akt2* does not significantly affect mTORC1 activity, as evidenced by S6K-pT389.

In addition, GSK3-S9 phosphorylation is governed by multiple other kinases in addition to Akt, including PKA (Li, et al. Cyclic AMP Promotes Neuronal Survival by Phosphorylation of Glycogen Synthase Kinase 3 β . *Molecular and Cellular Biology*. 2000. 9356-9363; Fang et al. Phosphorylation and inactivation of glycogen synthase kinase 3 by protein kinase A. *Proc Natl Acad Sci U S A*. 2000. 97(22):11960-11965), S6K1 (Zhang et al. S6K1 regulates GSK3 under conditions of mTOR-dependent feedback inhibition of Akt. *Mol Cell*. 2006. 24(2): 185–197) and RSK (Sutherland, et al. Inactivation of glycogen synthase kinase-3 β by phosphorylation: new kinase connections in insulin and growth-factor signaling. *Biochemical Journal*. 1993. 296 (1) 15-19; Sutherland, et al. The alpha-isoform of glycogen synthase kinase-3 from rabbit skeletal muscle is inactivated by p70 S6 kinase or MAP kinase-activated protein kinase-1 in vitro. *FEBS Letter*. 1994. 338(1):37-42). Moreover, GSK3-S9 phosphorylation can also be regulated by other GSK3 phosphorylation events such as GSK3-Y216 phosphorylation (Bhat et al. Regulation and localization of tyrosine²¹⁶ phosphorylation of glycogen synthase kinase-3 β in cellular and animal models of neuronal degeneration. *Proc Natl Acad Sci*. 2000. 97 (20) 11074-11079). Thus, GSK3-S9 phosphorylation may not always correlate with Akt-pT308 signals. In our study, GSK3-S9 phosphorylation is largely correlated with Akt-pT308 changes (revised Figs. 1h, 1j and 1i and revised Supplementary Figs. S3b and S3c).

• In some experiments, Akt protein levels are monitored by pan-Akt immunoblotting (e.g. 1g, 1j, 1l, etc.), whereas in other experiments, Akt expression is monitored by Akt1 immunoblotting (e.g. 1h, 2b, 2c, 2e, 2f, etc.). Moreover, Akt activation in some experiments is monitored via Thr308 phosphorylation analysis (e.g. in Fig 2k), whereas in other experiments this is done by analysis of phosphorylation at Ser 473 (e.g. Fig 2l), or both (e.g., Fig 2b, 2c). Authors should use uniform indicators of Akt expression/activity across experiments, or they should explain why they are using one specific marker in particular experimental settings.

Response: We thank the reviewer for bringing up these concerns. Following the reviewer's suggestion, we have rerun all lysates previously blotted with pan-Akt antibody and blotted again with anti-Akt1 antibody to keep all the Akt loading controls constant across the manuscript, including revised Figs. 1g, 1j, 1l, 3n, 4h and revised Supplementary Figs. S2b, S3a, S3b, S3c, S3d, S3e, S3f, S3h, S3i and S3m.

We also thank the reviewer for raising the concern regarding using different Akt phosphorylation antibodies in various experiments. As stated above, we mainly focus on understanding how SAV1 or MERTK manipulation affects Akt-pT308, given that pT308-Akt plays a relatively more important role in maintaining Akt kinase activity than pS473-Akt (Aoki et al. The Akt kinase: Molecular determinants of oncogenicity. *PNAS*. 1998. 95: 14950-14955; Rodrik-Outmezguine VS et al. mTOR kinase inhibition causes feedback-dependent biphasic regulation of AKT signaling. *Cancer Discovery*. 2011. 1: 248-59). In addition, we mainly used the Akt-pT308 antibody to examine whether certain cellular treatments work (such as PV treatment used in the revised Figs. 2i and 2k). As the reviewer kindly pointed out, the only figure panel in this manuscript we used Akt-pS473 but not Akt-pT308 as an indicator to examine whether indicated inhibitor treatments were functional or not, especially for PI3K inhibitor BKM120.

• The use of PV as a surrogate inducer of MERTK activity may be misleading, especially in respect to the effects on Akt Thr308 and Ser 473 phosphorylation. In fact, in Fig 2i, the PV-induced phosphorylation of Akt at Y26 is completely abrogated by a concomitant MERTK inhibition whereas phosphorylation at Thr308 is only partially inhibited. This indicates that PV has MERTK-independent effects on Akt phosphorylation. Similarly, PV induces Akt phosphorylation at Ser473 even in the presence of MERTK inhibitor that totally abrogates Akt phosphorylation at Y26 (Fig. 2l). Authors should comment on this.

Response: We fully agree with the reviewer that PV is not a specific MERTK upstream activator, rather PV treatment leads to activation of multiple RTKs, including but not limited to EGFR, VEGFR, PI3K and others. In our original manuscript, in addition to PV, we have also treated RCC4 cells with PtdSer to stimulate MERTK activation (Fig. 2h), in the presence of endogenous GAS6 (revised Supplementary figs. S6b and S6c). Moreover, as suggested by reviewer#2, we have also induced MERTK activation by GAS6, to specifically promote MERTK activation and subsequent Akt-T308 phosphorylation (revised Supplementary Fig. S6d). In original Fig. 2i (revised Fig. 2i), we fully agree with the reviewer that the PV-induced phosphorylation of Akt at Y26 is completely abrogated by a concomitant MERTK inhibition whereas phosphorylation at Thr308 is only partially inhibited. This data suggests that MERTK is the major kinase responsible for Akt1-Y26 phosphorylation, while Akt1-Y26 phosphorylation will facilitate subsequent Akt-T308 and S473 phosphorylation by PDK1 and mTORC2, respectively. However, in the absence of Akt1-Y26 phosphorylation (where SAV1 binds better to Akt), Akt-pT308 and pS473 can still occur but with a reduction (revised Figs. 2i, revised Supplementary figs. S6g, S6h, S12a and S12b). These data suggest that even without Akt1-Y26 phosphorylation and in the presence of SAV1 binding, Akt can be partially activated through canonical activating enzymes triggered by growth factor stimulations. This is consistent with our observations in original Fig. 2i that PV induced Akt-pT308 was reduced but not completely abolished upon MERTK inhibition by UNC2025.

As we have discussed in above responses, SAV1 mainly suppresses Akt-pT308 and only minimally reduces Akt-pS473 (revised Figs. 1j and 1l). Consistently, MERTK deletion or depletion also in large suppresses Akt-pT308 and only minimally reduces Akt-pS473 (revised Figs. 2b and 2c). Thus, our observation in Fig. 2l that inhibition of MERTK by UNC2025 moderately reduces Akt-pS473 signals, is consistent with MERTK genetic ablation and MERTK depletion data (revised Figs. 2b and 2c). Although there are crosstalks between Akt-pT308 and Akt-pS473, in this study we focus on examining roles of the MERTK/Akt/SAV1 signaling in regulating Akt-pT308 signals, which accounts for the majority of Akt kinase activity (Aoki et al. The Akt kinase: Molecular determinants of oncogenicity. *PNAS*. 1998. 95: 14950-14955; Rodrik-Outmezguine VS et al. mTOR kinase inhibition causes feedback-dependent biphasic regulation of AKT signaling. *Cancer Discovery*. 2011. 1: 248-59).

- **Which are the effects of the expression of a constitutively active MERTK mutant (Wu Y et al., Journal of Cell Science, 2005) on Akt phosphorylation (Y26, Thr308, Ser473) in response to Akt upstream agonists in RCC cell lines?**

Response: We thank the reviewer for bringing up this interesting question. Following the reviewer's suggestion, we obtained a constitutive-active MERTK from Dr. Raymond Birge group (Fc-MERTK, Kasikara et al. Phosphatidylserine Sensing by TAM Receptors Regulates AKT-Dependent Chemoresistance and PD-L1 Expression. *Molecular Cancer Research*. 2017. DOI: 10.1158). Notably, ectopic expression of Fc-MERTK in UMRC6 cells led to increased and sustained Akt-pT308 phosphorylation upon either EGF (revised Supplementary Fig. S6g) or insulin (revised Supplementary Fig. S6h) stimulation. These data suggest that MERTK facilitates Akt activation under both steady state conditions and acute EGF/insulin stimulation conditions.

- **Which are the roles of the SAV1-MERTK axis in respect to Akt signaling activation in cancer cells expressing constitutively active PI3K or EGFR mutants?**

Response: We thank the reviewer for raising this great suggestion to determine how MERTK/Akt/SAV1 signaling contributes to Akt activity control under constitutive PI3K or EGFR activation conditions. The PI3K signaling is moderately mutated but hyperactivated in RCC (Guo et al. The PI3K/AKT Pathway and Renal Cell Carcinoma. *J Genet Genomics*. 2015. 42(7): 343–353). EGFR mutation is not frequently observed in RCC patients (Bayrak et al. Evaluation of EGFR, KRAS and BRAF gene mutations in renal cell carcinoma. *J Kidney Cancer VHL*. 2014; 1(4): 40–45), while EGFR overexpression may associate with high tumor grade (Minner et al. Epidermal growth factor receptor protein expression and genomic alterations in renal cell carcinoma. *Cancer*. 2012. 118(5):1268-75).

We have searched for CCLE (cancer cell line encyclopedia) and cBioportal cancer cell line information and didn't find reported mutations in either PIK3CA or EGFR genes in RCC cell lines commonly used in labs for kidney cancer research, including 786-O, ACHN, UIMRC6, UMRC2, RCC4 and Caki-I cells. To evaluate effects of PIK3CA or EGFR hyper-activation in regulating the MERTK/Akt/SAV1 signaling, we ectopically expressed cancerous hot-spot hyperactivation mutations including PIK3CA-H1047R or EGFR-L858R, with their WT counterparts as controls in either SAV1-depleted RCC4 or Caki-I cells. We observed that ectopic expression of either PIK3CA-H1047R or EGFR-L858R led to increased basal Akt-T308 phosphorylation in RCC4 cells (revised Supplementary Figs. S10a to S10c). Moreover, ectopically expressed constitutively active PIK3CA and EGFR mutants could not compensate for the loss of MERTK in activating Akt (revised Supplementary Figs. S10b and S10c). Although depletion of endogenous SAV1 led to increased Akt-pT308 upon either EGF or insulin stimulation, expression of PIK3CA-H1047R in SAV1-depleted RCC cells (revised Supplementary Fig. S10d to S10f) did not significantly promote Akt-pT308 in responding to either EGF or insulin stimulation. Similarly, ectopic expression of EGFR-L858R also didn't result in elevated Akt-pT308 upon EGF stimulation (revised Supplementary Fig. S10i). To further examine whether PIK3CA or EGFR hyperactivation would compensate for the loss of MERTK, we ectopically expressed PIK3CA-H1047R or EGFR-L858R in MERTK-deleted RCC4 cells and found that hyperactivation of neither PI3K (revised Supplementary Figs. S10g and S10h) nor EGFR (revised Supplementary Figs. S10j) could largely rescue MERTK-deletion induced decreases in Akt-pT308. These data suggest that MERTK may be the major kinase through which SAV1-mediated Akt suppression is released.

In addition, from our signaling profiling of a panel of SAV1 expressing RCC cell lines (revised Supplementary Fig. S3m), RCC4 and Caki-I cells display relatively higher levels of Akt-pT308 than UMRC6 and UMRC2 cells. However, in both RCC cell lines with high Akt-pT308 basal levels (RCC4 and Caki-I) and low Akt-pT308 basal levels (UMRC6), depletion of endogenous SAV1 all led to elevation of Akt-pT308 (revised Figs.

1j, 1l and Supplementary Fig. S3a). These data suggest that SAV1-mediated Akt suppression mechanism may be functional for RCC cells in an Akt activation status-independent manner.

Minor points:

1) The quality of Akt-pThr308 immunoblotting of Fig 1h is very poor. The relevant bands have too much contrast and are half-visible. Please replace with better quality images.

Response: We sincerely apologize for the low-quality blot provided in previous figure panel. Following the reviewer's suggestion, we have replaced the image with a higher quality blot with a longer exposure.

2) In Fig S4, the labeling of the panel c is missing (there are two "b" labels).

Response: We sincerely apologize for mislabeling panel S4c and we have corrected it in the revised manuscript and figures.

3) The legend of panel f of Fig S8f is missing.

Response: We sincerely apologize for missing the legend for Supplemental Fig. S8f. We have included the legend for this Supplemental Figure panel in the revised manuscript.

REVIEWERS' COMMENTS:

Reviewer #1 (Remarks to the Author):

Much improved, accept.

Reviewer #2 (Remarks to the Author):

The authors have addressed my comments appropriately. The manuscript has improved strongly, and hence I would recommend this paper for publication.

Reviewer #3 (Remarks to the Author):

The authors have satisfactorily addressed all my comments/concerns with a significant amount of new experimental work that further corroborates the proposed mechanistic model. Therefore, I have no hesitation in recommending the manuscript for publication on Nature Communications.

Reviewer #1 (Remarks to the Author):

Much improved, accept.

Reviewer #2 (Remarks to the Author):

The authors have addressed my comments appropriately. The manuscript has improved strongly, and hence I would recommend this paper for publication.

Reviewer #3 (Remarks to the Author):

The authors have satisfactorily addressed all my comments/concerns with a significant amount of new experimental work that further corroborates the proposed mechanistic model.

Therefore, I have no hesitation in recommending the manuscript for publication on Nature Communications.

Response: We sincerely thank three reviewers in thoroughly evaluating our original and revised manuscripts. We also thank the reviewers for insightful and constructive suggestions that have been guiding us to further strengthen our manuscript during last round of revision.